# Persistent Parental RNAi in the Beetle *Tribolium castaneum* Involves Maternal Transmission of Long Double-Stranded RNA

*Thorsten Horn, Kalin D. Narov, and Kristen A. Panfilio\**

Parental RNA interference (pRNAi) is a powerful and widely used method for gene-specific knockdown. Yet in insects its efficacy varies between species, and how the systemic response is transmitted from mother to offspring remains elusive. Using the beetle *Tribolium castaneum*, an RT-qPCR strategy to distinguish the presence of double-stranded RNA (dsRNA) from endogenous mRNA is reported. It is found that injected dsRNA is directly transmitted into the egg and persists throughout embryogenesis. Despite this depletion of dsRNA from the mother, it is shown that strong pRNAi can persist for months before waning at strain-specific rates. In seeking the receptor proteins for cellular uptake of long dsRNA into the egg, a phylogenomics profiling approach of candidate proteins is also presented. A visualization strategy based on taxonomically hierarchical assessment of orthology clustering data to rapidly assess gene age and copy number changes, refined by sequence-based evidence, is demonstrated. Repeated losses of SID-1-like channel proteins in the arthropods, including wholesale loss in the Heteroptera (true bugs), which are nonetheless highly sensitive to pRNAi, are thereby documented. Overall, practical considerations for insect pRNAi against a backdrop of outstanding questions on the molecular mechanism of dsRNA transmission for long-term, systemic knockdown are elucidated.

## 1. Introduction

Since the demonstration of systemic RNA interference (RNAi) in insects about 20 years ago,[1–3] this technique has become widely used for genetics research and there is growing interest in its application for species- and gene-specific pest management.[4–9] In

T. Horn, K. A. Panfilio
Institute for Zoology: Developmental Biology
University of Cologne
Zülpicher Straße 47b, 50674 Cologne, Germany
E-mail: Kristen.Panfilio@alum.swarthmore.edu
K. D. Narov, K. A. Panfilio
School of Life Sciences
University of Warwick
Gibbet Hill Campus, Coventry CV4 7AL, UK

many species, systemic knockdown is efficient across life history stages, with a particular advantage of parental RNAi (pRNAi). Delivery of double-stranded RNA (dsRNA) into the mother, often by a single injection, can achieve knockdown of both maternal and zygotic gene expression in offspring, including at postembryonic stages.[10] This technique can provide highly efficient gene knockdown in hundreds of embryos that are often collected for up to 3 weeks after injection (e.g., refs. [1, 11]).

As a well-established model system, the red flour beetle, *Tribolium castaneum*, has been at the forefront of research on the RNAi mechanism[1,10,12] and for diverse genetics studies.[13] It is an effective RNAi screening platform.[14–16] pRNAi in *Tribolium* is regularly used for phenotypic investigation of development and to test genetic interactions singly or globally, such as by RNA-seq after RNAi.[17–20] Empirical work has shown that efficient RNAi is achieved through the introduction of long dsRNA into the organism, which persists longer in vivo and has more efficient cellular uptake than short interfering RNA (siRNA).[10,21] Supporting this, an early genomic survey of RNAi molecular machinery in *Tribolium* confirmed conservation of many core elements, but also with notable absences or changes in copy number or function of some elements compared to the well understood RNAi system of *C. elegans*.[12] This has generally been borne out by studies in other insect species.[4,5]

However, the mechanism of pRNAi is still poorly understood. Germline tissues and developing eggs have been studied as one of several tissue types that exhibit distinct susceptibilities to systemic knockdown in adult females. On the one hand, germline tissue showed lower levels of systemic effect in a pea aphid study in which this tissue was distal to the site of initial dsRNA delivery.[9] On the other hand, research in *C. elegans* has shown co-localization of dsRNA and yolk in oocytes, suggesting dsRNA transmission via a general mechanism for maternal provisioning of eggs.[22]

A key element for elucidating systemic pRNAi is the ability to detect and track the dsRNA. In *C. elegans*, microscopy for visual detection of fluorescently labeled dsRNA showed that

50-bp dsRNA was transmitted to the oocyte.[22] However, this qualitative study did not examine embryos beyond the four-cell stage or test long dsRNA (≈400 bp for efficient knockdown in *Tribolium*[10,15]). Visual tracking of fluorescently labeled dsRNA has been attempted in insects, but with limits on transmissibility and detection sensitivity.[23,24] Recent reviews on insect RNAi have thus explicitly called for the use of quantitative, sensitive detection methods such as RT-qPCR as a complementary approach: both to assay the extent of target gene knockdown after RNAi and for the systematic tracking of dsRNA.[6] RT-qPCR to assay knockdown is regularly used in developmental genetics research,[19,20,25] as one of several methods alongside global assays such as RNA-seq[17,20] and spatiotemporally sensitive methods such as in situ hybridization, which can also detect inter-embryo variability (e.g., ref. [25]). To the best of our knowledge, these methods have thus far been used to measure expression levels of endogenous target gene mRNA, but not for dsRNA detection.

Here, we combine experimental results in *Tribolium* with comparative genomics assessments of gene repertoires across species to shed further light on the molecular mechanisms of dsRNA transmission during systemic pRNAi in insects. We present an RT-qPCR strategy whose amplicon design and sensitivity distinguish dsRNA in offspring after pRNAi for genes with distinct temporal expression profiles, demonstrating its value for tracking throughout embryogenesis. Furthermore, we show that knockdown in progeny persists at high levels for months, despite a finite starting amount of dsRNA, through time-course analyses that evaluate female age, genetic strain, and different target genes. Finally, we compare hundreds of sequenced animal genomes to determine limits in the conservation of candidate receptor proteins for dsRNA uptake, showing the specificity of the importer protein SID-1 to nematodes compared to insects or vertebrates. Thus, even as we provide empirical advances for investigation and application of pRNAi, we also flag multiple aspects of dsRNA transport that remain enigmatic.

## 2. Results

### 2.1. dsRNA Is Transported into Eggs and Persists during Embryogenesis

The homeodomain transcription factor Tc-Zen1 is a critical regulator in early development, specifying the identity of the extraembryonic serosal tissue that surrounds the embryo and confers mechanical, physiological, and immunological protection.[18,20,26,27] During routine verification of *Tc-zen1* parental RNAi using RT-qPCR (as in ref. [20]), we unexpectedly found that measured expression of *Tc-zen1* was higher in RNAi samples than in wild type under certain assay conditions, despite strong phenotypic validation of systemic knockdown (see Experimental Section).

RT-qPCR is used to determine target gene expression levels after RNAi in terms of % of wild type (WT), calculated as $R_{RNAi}/R_{WT}$ (see Experimental Section). Higher RNA levels in an experimental condition (>100% of wild type) are interpreted as upregulation or overexpression, while lower levels (<100%) would imply downregulation or knockdown. If phenotypic assays document strong RNAi knockdown, why do RT-qPCR assays detect more target gene RNA after RNAi?

We observed this effect when using an RT-qPCR amplicon that was designed to be small and intron-spanning, ensuring efficient and specific amplification.[28,29] However, due to the small size of the *Tc-zen1* mRNA transcript, this amplicon was also nested within the region used as an established multi-purpose template for dsRNA and in situ hybridization (**Figure 1**A: Fragment 2, compared to the long dsRNA; refs. [20, 25, 30]). Using this amplicon, at young embryonic stages we observed strong reduction to 25% of wild type levels in the RNAi sample, consistent with our phenotypic validation (Figure 1B at 8–24 h: mean expression ratios of 1.24 RNAi/4.88 wild type for Fragment 2). In contrast, this amplicon produces higher expression estimates in RNAi than in wild type samples at older stages (Figure 1B: yellow versus red plot lines, developmental time ≥16–24 h). When the same samples are assayed with an RT-qPCR amplicon that only partially overlaps the dsRNA fragment (Figure 1A: Fragment 1), we obtain lower RNA levels in RNAi samples than wild type samples at all assayed stages (Figure 1B: blue plot lines), with RNAi knockdown to only 5% of wild type levels at 8–24 h (mean expression ratios of 0.22 RNAi/4.51 WT).

Notably, the semi-nested amplicon (Fragment 1) detects the same levels of wild type expression as our nested amplicon (Fragment 2; Figure 1B: light blue and red plot lines, respectively). This corroborates the accuracy of the initially used nested amplicon (Fragment 2) for quantification of *Tc-zen1* transcript levels. Moreover, these findings with either amplicon are consistent with our previous work that documented a single early pulse of *Tc-zen1* expression that peaks at 6–10 h before rapidly declining to undetectable levels for the rest of embryogenesis.[20]

Thus, we infer that after *Tc-zen1* RNAi the nested RT-qPCR amplicon is detecting both residual endogenous transcript and dsRNA transmitted from the mother to the egg. At older developmental stages when wild type expression is low or undetectable, the dsRNA would constitute the majority of all detected RNA. This is consistent with the observed higher levels of RNA in RNAi than wild type samples (Figure 1B: yellow versus red plot lines, developmental time ≥16–24 h). Under standard culturing conditions, *Tribolium* embryogenesis is ≈3 days, and here we show that the transmitted dsRNA stably persists in the egg throughout this interval (Figure 1B: yellow plot line, ≥16–24 h). Given phenotypic and molecular evidence (at 8–24 h) of RNAi knockdown, RNA degradation occurs in the embryo. Thus, our observation of stable dsRNA levels throughout embryogenesis suggests that dsRNA is transmitted into eggs at saturating levels that exceed our ability to detect a drop in dsRNA levels over time. Furthermore, although the nested fragment did capture the reduction in the target gene at a stage of high endogenous expression (8–24 h), the degree of transcript depletion after RNAi is likely underestimated due to the detection of the dsRNA (reduction to 25% with nested Fragment 2 versus to 5% with semi-nested Fragment 1). In summary, there is a certain amount of dsRNA transmitted from the mother to the offspring that is detectable by RT-qPCR, but at levels that may be overlooked at stages of high endogenous expression.

### 2.2. The Entire Long dsRNA Molecule Is Maternally Transmitted

The RNAi pathway involves processing of long dsRNA by the RNase III endonuclease Dicer to generate siRNAs of ≈20–23 bp,

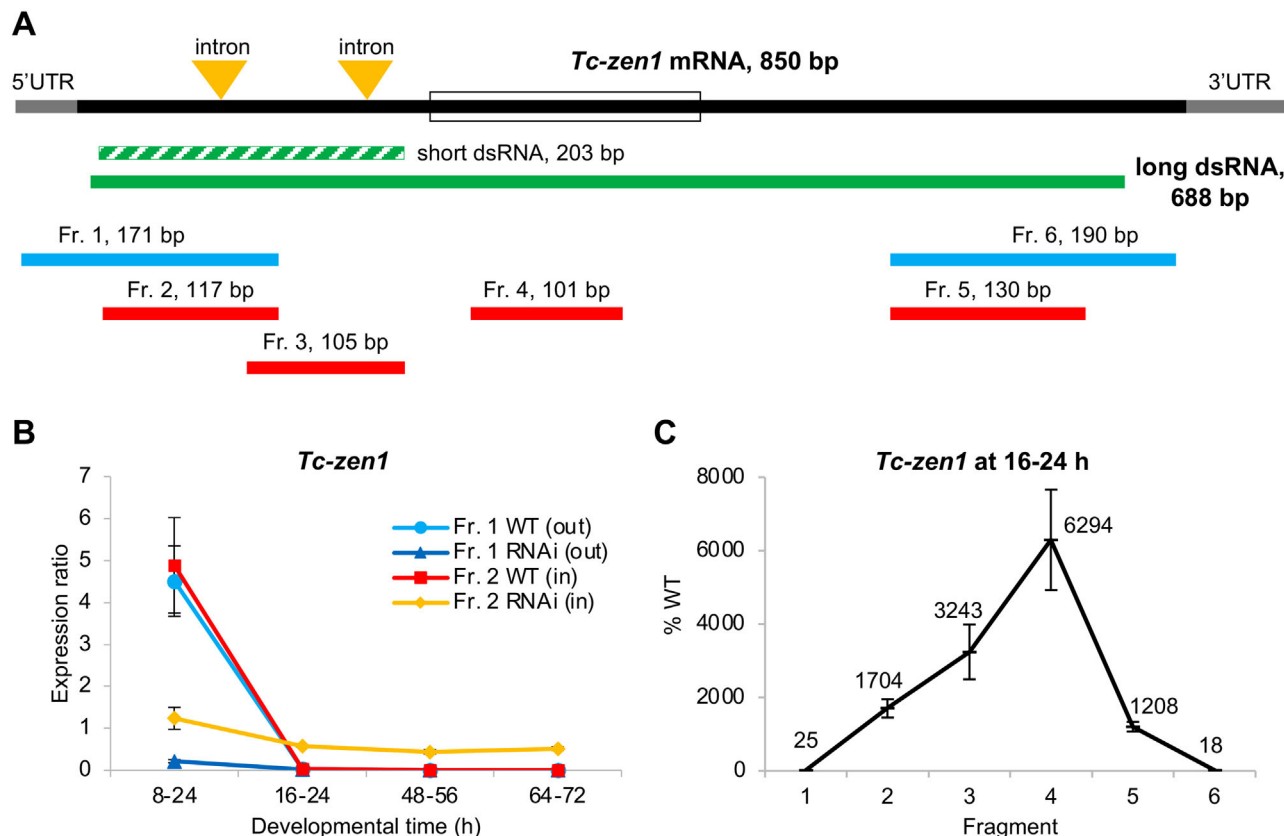

**Figure 1.** Long dsRNA molecules are transmitted maternally and persist throughout embryogenesis after parental RNAi for *Tc-zen1*. A) Structure of *Tc-zen1* mRNA (CDS: solid black, UTRs: grey, homeobox: open box) and corresponding dsRNA fragments (green) used to silence the gene: the long dsRNA (solid green) was used in this study; the short dsRNA (dashed green) was used previously[20] to specifically avoid the highly conserved homeobox. Beneath, the six fragments (Fr. 1–6) indicate the regions used for RT-qPCR quantification, where the two outermost fragments (blue) lay partially outside of the dsRNA fragment and four fragments (red) lay inside the dsRNA fragment. Fragment lengths are indicated and are shown to scale. B) Expression ratio of *Tc-zen1* in knockdown (RNAi) and wild type (WT) samples at different stages of development, assayed by RT-qPCR with fragments that extend outside (Fr. 1) or are nested within (Fr. 2) the dsRNA fragment, as indicated in the legend. In the three older stages, Fragment 2 in the RNAi samples (yellow) shows consistently higher expression than all other samples, due to its ability to detect the dsRNA in addition to endogenous transcript. Developmental time is specified in hours after egg lay (i.e., after fertilization). C) *Tc-zen1* expression measured by RT-qPCR in the RNAi samples compared to WT samples for all fragments, at a developmental stage when endogenous mRNA levels are very low (at 16–24 h). The two outermost fragments (1 and 6) show reduced expression compared to WT, consistent with successful RNAi knockdown, while the inner fragments (2–5) show increased expression after RNAi, with highest expression for Fragment 4 (see also Figure S2, Supporting Information). The mean values (%) for each fragment are indicated. Mean expression levels are shown from three biological replicates (see Experimental Section); error bars represent ± one standard deviation.

which is the means of amplifying the RNAi effect to systemic levels.[31] Yet, our nested RT-qPCR amplicon is >100 bp. We thus considered the possibility that the dsRNA is transmitted from the injected mother to the embryo as a largely intact, unprocessed molecule.

Our method to detect transmitted dsRNA relies on measuring different expression levels in the same sample with two different amplicons, one being partially outside of the dsRNA sequence. In theory, this method could also be used to determine the size of the transmitted dsRNA by increasing the length of the amplicons (e.g., by extending Fragments 1 and 2 in the 3′ direction). Unfortunately, RT-qPCR analysis becomes increasingly unreliable with increasing amplicon size,[29] and our results were inconclusive between biological and technical replicates with this strategy.

As an alternative approach, we could robustly measure the relative expression of a series of RT-qPCR amplicons that span the *Tc-zen1* transcript (Figure 1A: Fragments 1–6). As wild type ex-

pression is very low at 16–24 h (Figure 1B), the measured expression at this stage largely represents transmitted dsRNA present in the egg. Validating RNAi efficiency, the two amplicons that lay partially outside the dsRNA region show efficient knockdown of *Tc-zen1* at 16–24 h (Figure 1C: Fragments 1 and 6, mean reductions to ≤25% of WT levels). This is consistent with phenotypic validation and RT-qPCR assays of early developmental samples with high wild type expression (Figure 1B: 8–24 h). In contrast, all amplicons that were fully nested within the dsRNA region show substantially increased expression after RNAi (>1000%; Figure 1C: Fragments 2–5). Strikingly, there was a five-fold range in expression levels among the nested amplicons, an issue we address in the Discussion in terms of experimental design and gene-specific sequence features. Regardless, these four amplicons are each >100 bp and together span 654 bp. We thus conclude that the entire 688-bp dsRNA molecule injected into the mother is transmitted to the egg.

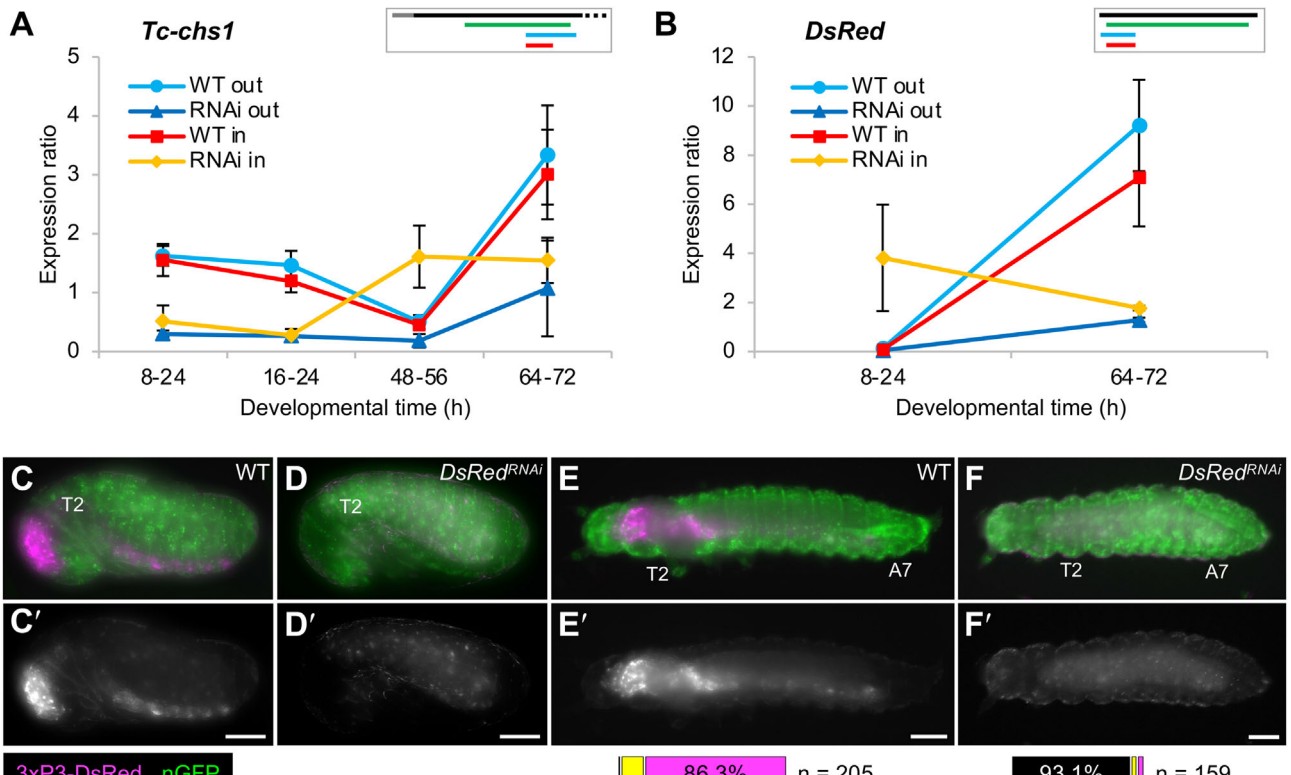

**Figure 2.** Maternal transmission of dsRNA occurs for diverse genes with distinct expression profiles. A,B) RT-qPCR expression ratio assayed with amplicons that are nested ("in": red and yellow) or partially outside ("out": light and dark blue) with respect to the dsRNA fragment, in wild type and after RNAi, as indicated in the legends. Mean expression levels are shown from three biological replicates; error bars represent ± one standard deviation. For *Tc-chs1* (A), the nested qPCR amplicon shows higher expression in RNAi samples (yellow) when endogenous *Tc-chs1* expression is low (48–56 h). Similarly, in the nGFP strain expressing transgenic dsRed (B), the *DsRed* nested qPCR amplicon detects more RNA in the RNAi than wild type samples at a stage when the *DsRed* transgene is not expressed (8–24 h). Inset schematics depict the transcript, dsRNA, and qPCR fragments to scale, using the same color scheme as in Figure 1; only the first 700 bp of the 5092-bp mRNA is shown for *Tc-chs1*. C–F) Phenotypic confirmation of *DsRed* knockdown through loss of DsRed fluorescence in a transgenic line that ubiquitously expresses nuclear-localized GFP (green). The 3xP3 core promoter drives DsRed signal (magenta) in the brain and ventral nerve cord of untreated control (WT) embryos (C) and larvae (E). After *DsRed* RNAi, 3xP3-driven DsRed signal is absent, with only weak autofluorescence detected in the epidermal cuticle and the yolk (D,F). Views are lateral (C,D) or dorsal (E,F), with anterior left and, as applicable, dorsal up. Landmark thoracic (T) and abdominal (A) segments are numbered. Letter-prime panels show the DsRed channel alone. Scale bars are 100 µm. Horizontal bar charts show the proportions of larvae with no (black), weak (yellow), or strong (magenta) DsRed signal in larvae (*n* = 205 for WT, *n* = 159 for RNAi).

## 2.3. dsRNA Detection at Stages of Low Expression Is a General Feature

We next expanded our analyses to test whether maternal dsRNA transmission is a general feature of systemic RNAi in *Tribolium*. For this purpose, we chose two additional genes that have distinct, well-characterized expression time courses and molecular functions that differ from *Tc-zen1* and from one another. The first gene, *Tc-chitin synthase 1* (*Tc-chs1*), encodes a large, transmembrane enzyme that extrudes the polysaccharide chitin into developing cuticle of the serosa (early embryogenesis[27]) and of the larval epidermis (late embryogenesis[32]). Second, in the nuclear GFP (nGFP) line,[33] red fluorescence encoded by *DsRed* serves as a transgenic marker under the control of the synthetic Pax6 core promoter-enhancer element 3xP3, which drives late expression in the developing eyes and ventral nerve cord (central nervous system[34,35]).

For both genes we detected greater expression in the RNAi samples with the nested amplicon compared to the semi-nested amplicon (**Figure 2**A,B: yellow versus dark blue plot lines). Furthermore, the effect was again most pronounced—with more RNA in the RNAi than wild type samples—at developmental stages when wild type expression is low: early embryogenesis for *DsRed* (4733%) and mid-embryogenesis for *Tc-chs1* (322%). As we had observed this effect in late embryogenesis for *Tc-zen1* (Figure 1B), these results clarify that it is the level of endogenous expression, and not a specific developmental stage, that determines when dsRNA transmission can be strongly detected with our RT-qPCR strategy. This is applicable whether the gene has a single stage of peak expression (*Tc-zen1, DsRed*) or a bimodal temporal expression profile with only a transient period of low expression (*Tc-chs1*). At stages when the target gene is moderately to strongly expressed, for both *Tc-chs1* and 3xP3-driven *DsRed* the nested amplicon underestimates the level of knockdown

after RNAi by 5–20%, similar to what we had observed for *Tc-zen1*.

We also verified the knockdown efficiency for *DsRed* in the nGFP line by observing red fluorescence in late embryos and young larvae (Figure 2C–F). Fluorescent signal was detectable in >99% of untreated (wild type) larvae ($n = 205$) and absent in 93.1% of RNAi larvae ($n = 159$), consistent with very high efficiency knockdown.

### 2.4. pRNAi Is Highly Efficient for Months before Waning at Strain-Specific Rates

A single injection of the mother provides a finite number of dsRNA molecules, and the knockdown effect of pRNAi wanes over time in insects.[1,3,36] Our results suggest that waning may reflect not only endogenous transcript recovery after dsRNA degradation in the mother, but also maternal depletion of dsRNA due to its direct transmission into offspring. To determine how long pRNAi knockdown persists in *Tribolium*, we conducted time course experiments until the knockdown effect had fully waned, testing different genes, genetic backgrounds, and ages of adult female. For this purpose, larval cuticle preparations were used as a robust phenotype assay (see Experimental Section), targeting two genes whose knockdown produces distinctive and easily scorable cuticle phenotypes with high penetrance (**Figure** 3A–C): *Tc-tailup* (*Tc-tup*[15,37,38]) and *Tc-germ cell-less* (*Tc-gcl*[39]).

Across beetle strains and target genes, >90% penetrance for gene-specific knockdown in embryos is achieved within 3 days after adult injection and remains persistently high for nearly 2 months at 30 °C (Figure 3D: Experiments 1, 3a, and 3b). Only in our aged female experiment did we see a delay in onset of knockdown and lower overall levels of penetrance (generally 50% over a 30-day interval; Figure 3D: Experiment 2). Nonetheless, across all experiments we still observed 50% phenotype penetrance at 42–71 days after injection. A minor resurgence (<10%) after full depletion of the RNAi phenotype occurred briefly toward the end of both Experiments 2 and 3a.

In contrast to the consistent duration of strong knockdown, the rate of waning may be strain-specific, irrespective of female age or target gene. In Strain 1, knockdown fully declined in a 10-day interval (from 91% or 78% to 0% in Experiments 1 and 2, respectively). Waning in Strain 2 was more gradual, spanning the better part of a month (from ≈86% to 0% over 20–34 days in Experiments 3a and 3b).

### 2.5. pRNAi Waning and Transient Fluctuations Are Strain- and Female-Specific

Since our experimental beetle populations were maintained as pooled cohorts, we examined female lethality and fecundity to more precisely document the pRNAi waning effect (Figure 3E–H).

Regarding survival (Figure 3E), the dsRNA-injected females exhibited minor fatalities within the first week after injection before the populations stabilized over the next 1–2 months, until death occurred from presumed old age. The exception to this trend was in Experiment 2, where females were already aged

for 5.3 months as adults before injection and subsequent mating: these injected females showed steady mortality for the first 2.5 weeks before the population stabilized through the second month of the experiment. Fatalities of the uninjected (wild type) females and males were minimal in all experiments.

We then determined fecundity in terms of egg output per female per day (Figure 3F). Age is the strongest predictor of female fecundity; neither the background genetic strain nor dsRNA injection had an appreciable effect. Fecundity fluctuates on short time scales (<1 week), but overall we find a marked but inexplicable increase in fecundity at 50–75 days, with ≥6 eggs/female/day. After, there is a rapid decline to 130 days, and persistent, low-level fecundity through 230 days.

In sum, we find that on multi-month timescales both survival and egg output of RNAi females is comparable to that of the uninjected controls, indicating that long-term activity of RNAi machinery does not generally impair female physiology or fecundity.

Arguably, intermediate RNAi penetrance at the population level could reflect offspring contributions from a mix of females with strong RNAi and resistant females that only lay wild type offspring. Then, waning of RNAi over time might reflect the earlier death of the females that produced affected offspring. However, our data support the waning of RNAi in individual females. First, for months we obtained exclusively affected offspring (100% RNAi phenotype) before eventually obtaining 0% phenotype (Figure 3D: Experiments 1, 3a, and 3b). Second, RNAi penetrance fluctuates and wanes even when the number of females and egg laying rate are steady (Figure 3G,H). Thus, while we cannot formally exclude individual differences in reproductive senescence,[40] decline in RNAi penetrance was not simply due to death of females in which RNAi was more effective.

### 2.6. Multiple, Independent Losses of the dsRNA Importer SID-1 in Arthropods

For the transit of dsRNA through the mother to the egg, diverse receptor proteins have been implicated in dsRNA cellular uptake and oocyte provisioning. In widening our investigation of the molecular mechanisms of pRNAi, we took a phylogenomic approach to explore the potential relevance of selected receptor proteins in insects. Moreover, our analyses demonstrate a systematic approach for conservation assessments that combines extensive orthology clustering datasets with curation and phylogenetic analysis.

RNAi requires that dsRNA is taken up into the cells of the body, where Dicer acts in the cytosol.[4,5] The SID-1 protein is a transmembrane importer of long dsRNA and has been a central focus of RNAi research. First characterized in *C. elegans*,[41] it is one of several functionally related proteins whose absence causes a systemic RNA interference deficient (SID) phenotype (reviewed in refs. [42, 43]). Conservation of SID-1 is in fact notably variable across insect species, with homologs somewhat agnostically referred to as SID-1-like (SIL) or SID-1-related (Sir).[12] Nonetheless, ever since early recognition of SID-1 homologs in *Tribolium* and vertebrates,[41] it is routinely sought when characterizing RNAi components in new transcriptomes and genomes (see Discussion).

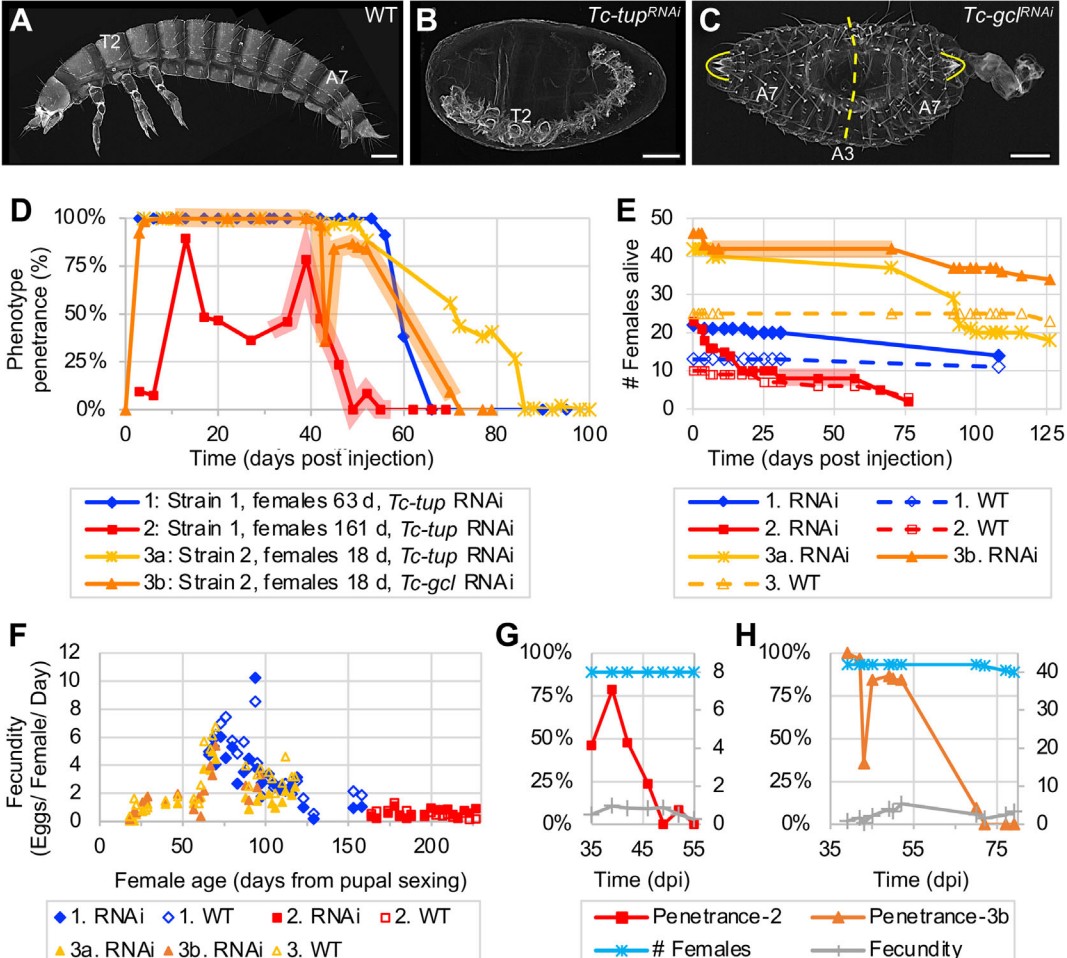

**Figure 3.** Systemic parental RNAi persists at high levels for months before fully waning. A–C) Representative larval cuticle preparations for wild type (WT), *Tc-tup*^RNAi^, and *Tc-gcl*^RNAi^ (from Experiment 3, collected 39–52 dpi, assayed ≥6 days after egg lay). Views are lateral (A,B) or dorsal-lateral (C), with anterior left and dorsal up. Landmark thoracic (T) and abdominal (A) segments are numbered. The dashed line indicates the plane of symmetry in the *Tc-gcl*^RNAi^ mirror-image double abdomen phenotype; brackets outline the terminal urogomphi. Scale bars are 100 µm. D) Time courses of parental RNAi penetrance from experiments that differ in beetle strain, female age, and target gene for knockdown (see figure legend and Experimental Section). Data points represent minimum age after injection, with n ≥ 10 eggs in each sample (see Experimental Section). Shaded plot segments for Experiments 2 and 3b represent time intervals with dynamic changes in RNAi penetrance that encompass both transient fluctuations (increase or decrease) and the interval of RNAi waning, while female population size was constant (no fatalities). E) Survival curves for females from all treatment conditions from all three experiments. For Experiments 2 and 3b, respectively, the red and orange shading corresponds to the same intervals as in (A). F) Fecundity values (number of eggs per female per day) relative to female age from all treatment conditions in all experiments, assayed at 19–26 time points per treatment. G,H) Juxtaposition of phenotype penetrance (%, left *y*-axis) with female population size and fecundity values (integer values, right *y*-axis) for the period of RNAi waning in Experiments 2 and 3b (red and orange shaded intervals, as above): female population size and fecundity remain steady or exhibit only minor fluctuation while RNAi wanes.

In the last 5 years the substantial increase in available genomic resources, particularly for the wider diversity of insects,[44] enables a more systematic approach based on official gene set (OGS) data from sequenced genomes. Here, we make use of the latest version of the orthology clustering database OrthoDB to survey 148 insect species, embedded in the evolutionary framework of 448 metazoan animal species (ref. [45]; **Figure 4**: cladogram).

Our assessments of orthology group membership at the hierarchical taxonomic levels of Insecta, Hexapoda, Arthropoda, and Metazoa substantially extend previous observations on the distribution of SID-1 (Figure 4: "SID-1/SIL distribution"; see Ex-

perimental Section and Discussion). Across the Metazoa, SID-1 proteins are present in 375 species, with multiple copies found in 235 of these species. As previously documented with limited sampling,[12] we find lineage-wide copy number increases within each of the sarcopterygian vertebrates (the lobe-finned fishes clade, including mammals), Coleoptera (beetles), and Lepidoptera (moths and butterflies). This includes the three SIL proteins originally characterized in *Tribolium*.[12] At the same time, SID-1 is absent from all 56 species of Diptera and 7 Acari species, augmenting previous reports.[46,47] Furthermore, we newly report the complete absence of SID-1 homologs in an additional, independent lineage: the Heteroptera (true bugs)

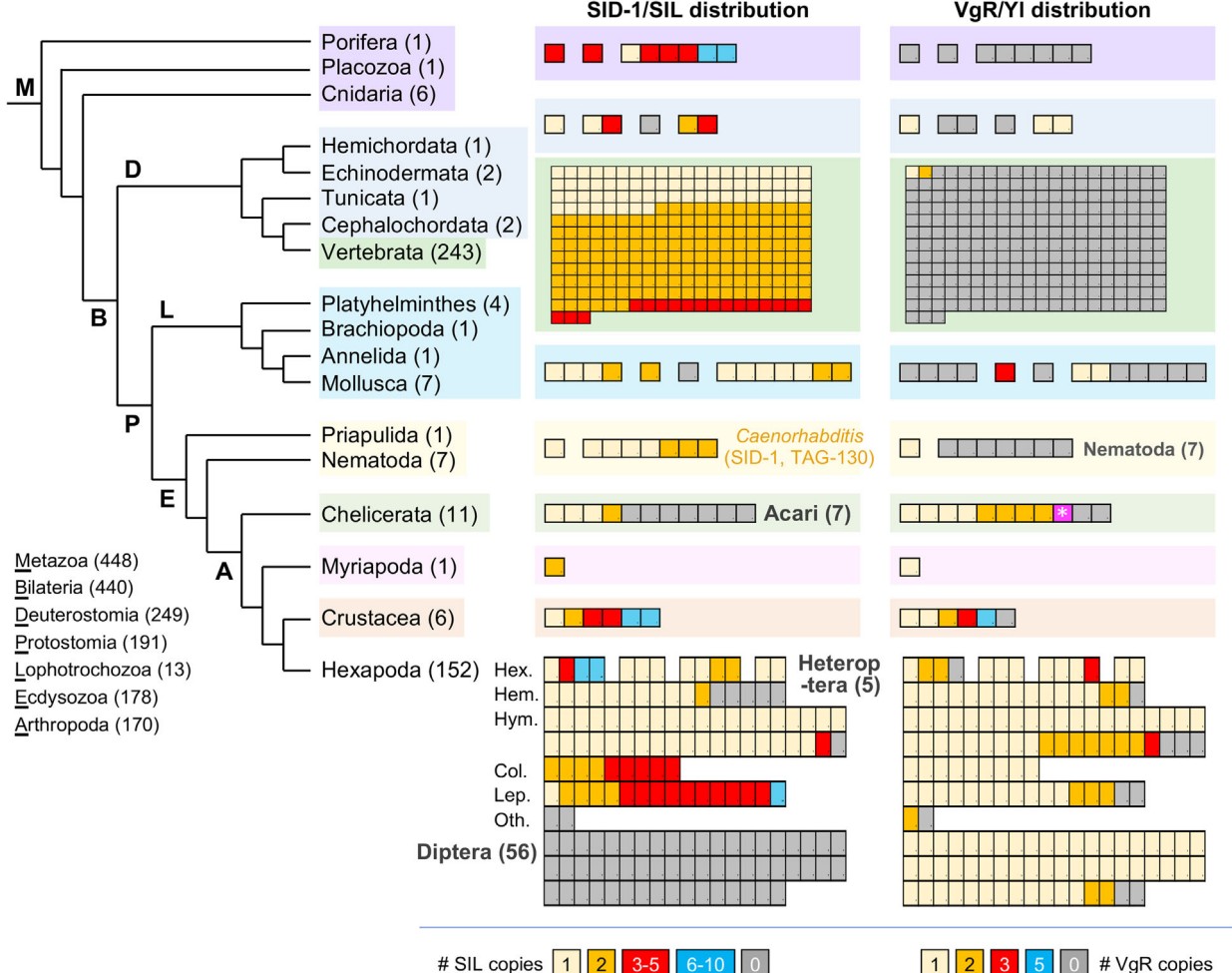

**Figure 4.** Visualization of metazoan orthology clustering reveals macroevolutionary patterns of protein conservation and lineage-specific losses. Taxonomic distribution and copy number of the SID-1/SIL and VgR transmembrane receptor proteins, representing all metazoan animal species in OrthoDB v10.1, with species numbers stated parenthetically. Phylogenetic relationships are based on refs. [87, 88]. Protein distributions are shown with one box per species, ordered sequentially by copy number, with the color code indicated in the legend for each gene. Notable lineage-specific absences are indicated in bold grey text. For one mite species (Acari), a VgR protein was only included in the wider metazoan orthology group, but this species did not have a VgR protein based on orthology clustering of Arthropoda only (magenta with white asterisk). No other presence/absence results differed across the Insecta, Hexapoda, Arthropoda, and Metazoa clustering analyses. For minor changes in copy number across clustering analyses, the value reported here is based on the most taxonomically restricted analysis (see Experimental Section). Hexapoda taxonomic abbreviations and species counts: Hex.: Non-insect Hexapoda (4), Palaeoptera (3), Polyneoptera (4), Non-hemipteran Paraneoptera (2); Hem.: Hemiptera (16); Hym.: Hymenoptera (40); Col.: Coleoptera (9); Lep.: Lepidoptera (16); Oth.: other Holometabola: Strepsiptera (1), Trichoptera (1). Vertebrate SID-1 proteins are mostly multi-copy, with single orthologs in ray-finned fishes (Actinopterygii), some orders of birds (Pelecaniformes, Gruiformes), and the platypus.

within the insect order Hemiptera (Figure 4). To corroborate these evolutionary changes, we further scrutinized OGS, genome assembly, and transcriptome analysis data.

Orthology clustering indicates the lineage-specific loss of SID-1 within the Hemiptera based on its absence in 5 Heteroptera and presence in 11 outgroup species (formerly the paraphyletic "Homoptera," including aphids, psyllids, and planthoppers; Figure 4). To augment species sampling, we compiled recently published results and conducted BLAST investigations of assembled genomes (see Experimental Section), nearly doubling the number of species investigated (**Figure 5**A). Importantly, directly interrogating genome assemblies overcomes limitations of OGS gene model predictions.[48,49] Our tBLASTn searches

with diverse SIL ortholog queries did not detect any heteropteran or dipteran sequences but did recover all SIL proteins in other insects (Figure 5B). Thus, loss of SID-1/SIL spans the 4 major infraorders of Heteroptera (10 species) compared to its retention in other Hemiptera (present in 15 species, with absences confined to 3 taxonomically scattered species with limited transcriptomic evidence; Figure 5A).

Even with more extensive species sampling than was previously possible,[12,46] some of the same phylogenetic ambiguities of SIL proteins remain (Figure 5C; Figure S1, Supporting Information). Within *Caenorhabditis* nematodes, SID-1 has high sequence similarity to the functionally unrelated TAG-130/CHUP-1 protein (Figures 4 and 5C; ref. [12]). Our phyloge-

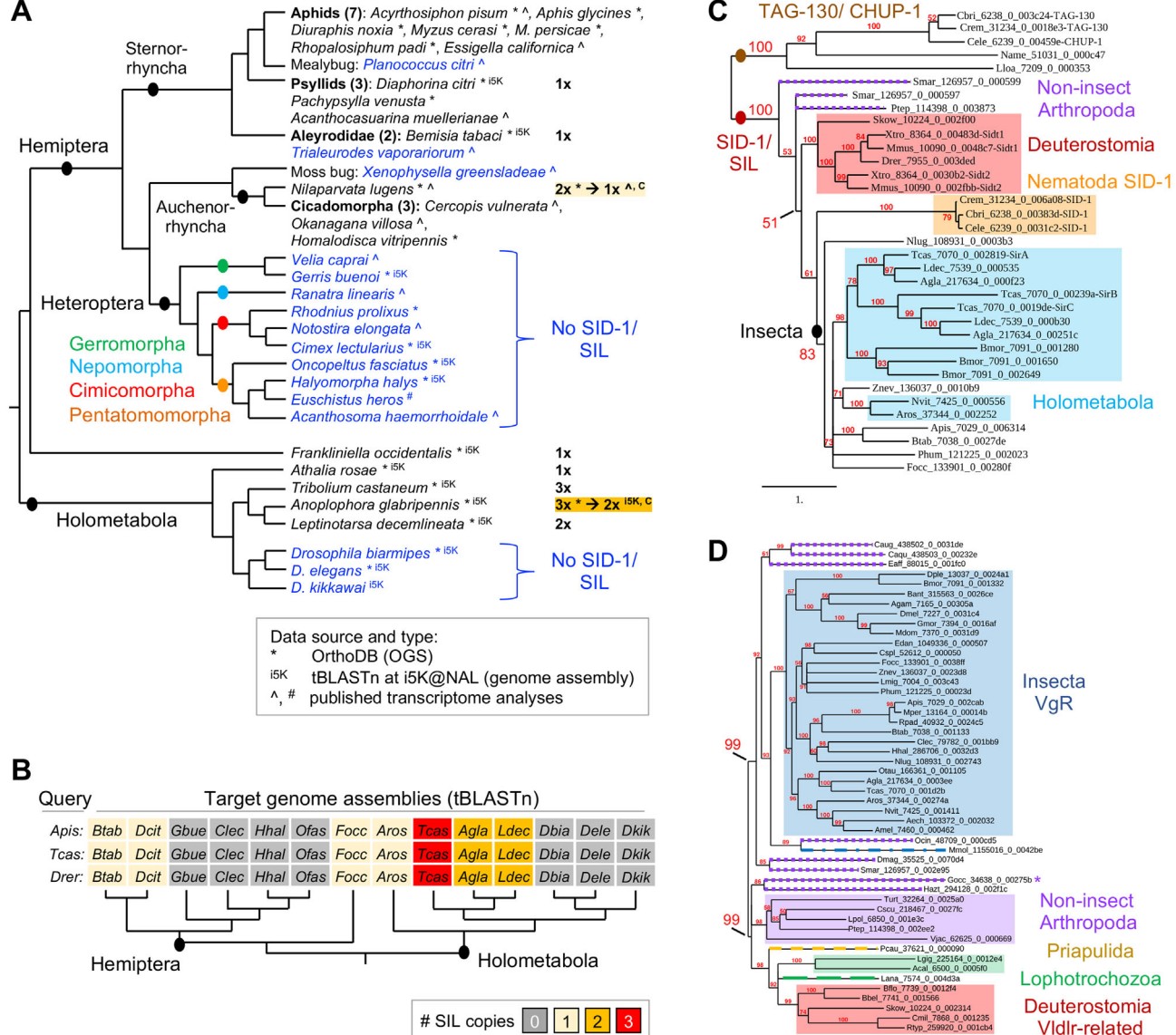

**Figure 5.** Curation, BLAST, and phylogenetics confirm and refine orthology clustering assessments of SID-1 and VgR distributions. A) Detailed evaluation of genomic resources for Hemiptera and selected outgroups supports the lineage-specific loss of SID-1 in the Heteroptera: species in blue text lack SID-1. Data types and sources are indicated in the legend, including recent transcriptomes (^: ref. [46]; #: ref. [7]), genome assemblies (i5K: ref. [78]), and OGS collections at OrthoDB (*: ref. [45]). Phylogenetic relationships after refs. [88–90]. For two species (*Nilaparvata lugens* and *Anoplophora glabripennis*), follow-up curation ("C") reduced SID-1 copy number compared to the OrthoDB assessment, as indicated (see Experimental Section). B) Selected subset of 14 species from (A) that were further interrogated by direct tBLASTn searching of the genome assembly. Each of the three orthologous query proteins from *A. pisum*, *T. castaneum* (SirA), and *Danio rerio* produced identical outcomes for copy number. C,D) Maximum likelihood whole-protein phylogenies of SID-1 homologs based on 35 proteins from 23 species (C) and VgR/Vldlr homologs based on 50 proteins from 50 species (D). The branch length unit represents substitutions per site. All nodes have ≥50% support (enlarged labels for selected nodes). Shaded boxes indicate clades of interest, as labeled in the figure, with dashed colored lines for paraphyletic protein members. For the VgR/Vldlr tree, the protein marked with an asterisk (*) represents the chelicerate species that was only included in the Metazoa, but not the Arthropoda, orthology clustering analysis (see Figure 4).

nies are generally robust for topology within clades for the insects and the deuterostomes, but the long-branch nematode proteins are unstable. Two nematode species with single-copy orthologs have particularly long branches and tend to show affinity with *Caenorhabditis* TAG-130. However, the recovery of well-supported clades for each of SID-1 and TAG-130 in *Caenorhabditis* species is inconsistent (Figure S1A–C, Supporting Information). In our phylogeny with broad species sampling, all arthropod and deuterostome proteins show greater affinity to nematode SID-1 (Figure 5C). Lineage-specific duplications appear ancestral, with a single duplication at the base of the sarcopterygian vertebrates and the beetles, and two at the base of the Lepidoptera (Figure 5C; Figure S1B,D, Supporting Information), but with unstable topology for *Tribolium* SirB. The Hymenoptera (wasps, bees) are an

outgroup to other Holometabola, yet their single-copy SIL orthologs group elsewhere (Figure 5C; Figure S1D, Supporting Information). Overall, sequence-based assessments of SID-1/SIL conservation are complicated by lineage-specific duplications and rates of sequence evolution, even before its functional relevance for RNAi in insects is considered (see Discussion).

### 2.7. Maternal Provisioning Uses Distinct Receptor Proteins in Insects and Nematodes

An alternative, long-recognized mechanism of dsRNA cellular uptake is endocytosis, for which core genes are widely conserved as standard eukaryotic cellular machinery.[4,42] Receptor-mediated endocytosis also supports maternal provisioning of oocytes, and it has been proposed for invertebrates that yolk proteins (vitellogenins) and dsRNA may share a common import mechanism.[22,23] We thus applied our phylogenomic approach to determine conservation of the vitellogenin receptor (VgR), known as Yolkless (Yl) in *Drosophila* (Figures 4 and 5D).

We find a fundamentally different distribution for VgR compared to SID-1 (Figure 4: "VgR/Yl distribution"). Whereas SID-1 had orthology group members extending to the non-bilaterian Metazoa, VgR is essentially restricted to the Ecdysozoa, excluding the Nematoda. Second, whereas there is evidence for multiple VgR proteins in other arthropod groups, this protein is predominantly single-copy throughout the insects, including the Heteroptera and Diptera, and the Coleoptera and Lepidoptera—which lost or duplicated SID-1, respectively. Unlike SID-1, for VgR there are also scattered single-species absences throughout the hexapod orders.

Curiously, two species are the sole exception to the complete absence of vertebrate protein members from the metazoan VgR orthology group (Figure 4). Our phylogenetic appraisal centered on this anomaly. We obtain two strongly supported clades containing either insect VgR or the deuterostome proteins, with a paraphyletic splitting of non-insect arthropod proteins between these two clades (Figure 5D). Tracking the vertebrate proteins into the more taxonomically restricted Vertebrata orthology group revealed that these proteins are divergent members of the Very Low-Density Lipoprotein Receptor (Vldlr) proteins, which are conserved in all 243 vertebrate species. In summary, the broad distribution patterns suggested by orthology clustering alone are valid, with our follow-up analyses refining this to strongly support a hexapod-specific origin of VgR. Thus, for the purposes of maternal provisioning of oocytes, nematodes and insects rely on distinct receptors.

## 3. Discussion

Our tripartite investigation of the molecular mechanism of pRNAi in *Tribolium* combines 1) an RT-qPCR strategy that detects dsRNA transmitted to the egg, 2) time course assays that show months-long persistence of pRNAi under different parameters, and 3) a phylogenomics profiling approach for appraisal of candidate genes' taxonomic distributions. Our surprising empirical observations can inform experimental design for developmental genetics studies and targeting strategies for RNAi-based

pest management applications. Furthermore, we highlight several key steps at which the cellular mechanism of dsRNA transport remains unresolved, despite highly effective use of RNAi in insects for decades.[1,2,5,6,15]

### 3.1. Amplicon Design and Developmental Staging Determine Measured Knockdown Efficiency

We show that comparison of RT-qPCR results between nested and semi-nested amplicons is a robust method for detection of maternally transmitted dsRNA in eggs (Figures 1 and 2). Complementing short-term tracking of fluorescently labeled dsRNA,[6,22,23] our method detects dsRNA throughout embryogenesis. On the other hand, use of a nested amplicon alone may lead to underestimation of knockdown efficiency, or even to erroneous interpretations of target gene overexpression, depending on endogenous expression levels. Awareness of these features can be applied to tracking dsRNA and to mitigate against unwanted dsRNA detection in single-amplicon assays.

For a gene of interest, primer design may be constrained such that an RT-qPCR amplicon is nested within the dsRNA region. For example, to design intron-spanning primers for short, efficient amplicon sizes,[28,29] while also avoiding conserved coding sequence regions that could cause off-target effects,[15,20] both RT-qPCR and dsRNA primers may target the same region. Second, small genes with few introns are particularly constrained, such as *Tc-zen1* (Figure 1A: Fragment 3 with respect to the short dsRNA that avoids the homeobox, as in ref. [20]). Third, for efficient screening of both expression and function, a single longer amplicon may serve as template for both in situ hybridization, where probe sensitivity correlates with sequence length,[50] and for RNAi, where longer dsRNA is more effective.[10] This is the case with the long dsRNA for *Tc-zen1* examined here (Figure 1A; ref. [25]).

We find that nested amplicons underestimate true knockdown strength by 5–20% compared to measurements with semi-nested amplicons that only detect endogenous transcript (Figures 1B and 2A,B). Yet in previous work we consistently obtained strong knockdown validation with a nested amplicon, to 10% of wild type levels (ref. [20]: Fragment 3 and the short dsRNA, Figure 1A). A key factor was tight developmental staging that targeted peak endogenous expression. Broad sampling beyond the peak expression window effectively dilutes the detection of wild type endogenous transcript levels as the baseline against which RNAi samples are compared. This can substantially alter calculations of knockdown efficiency (**Figure 6**). More generally, knockdown assays based on developmental stages of low endogenous expression are less sensitive, even with semi-nested or un-nested amplicons that strictly detect endogenous mRNA. Calculations of RNAi knockdown levels are based on the ratio of target gene RNA in RNAi versus control (wild type) samples. Thus, when wild type levels are low, the numerical range—the sensitivity—of the denominator is reduced for $R_{RNAi}/R_{WT}$. For example, for *Tc-chs1* we obtained two-fold variation in calculated knockdown level from different developmental stages of the same experiment, with either nested or semi-nested amplicons (Figure 2A). Thus, staging precision is critical for accurate detection of both wild type expression levels and RNAi knockdown efficiency, and this

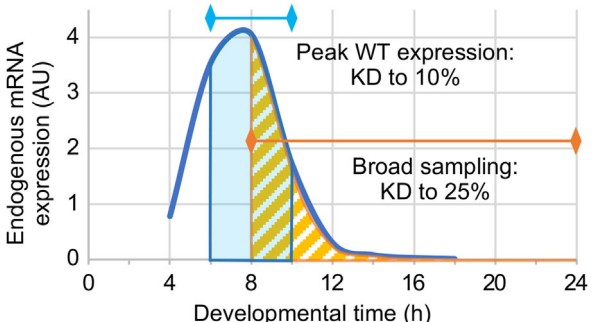

**Figure 6.** Tighter developmental staging mitigates underestimation of RNAi knockdown when assayed with a nested qPCR amplicon. This schematized representation based on empirical data for *Tc-zen1* illustrates how the time window assayed by RT-qPCR compares to the time course of endogenous expression,[20] and in turn how this affects the apparent efficiency of RNAi knockdown (KD). Even with a nested amplicon, which detects both endogenous mRNA and dsRNA, assays that strictly target the time window of peak endogenous expression document strong RNAi knockdown (blue: based on use of Fragment 3 depicted in Figure 1A; ref. [20]). In contrast, broad sampling that includes periods of low endogenous expression is more susceptible to underestimation of knockdown (orange: to 25% of wild type levels, based on Fragment 2, data in Figure 1B). This is because total RNA from broadly staged samples has a higher proportion of dsRNA relative to endogenous mRNA.

can largely overcome the underestimation effect of using a nested amplicon.

Measured expression levels are also affected by sequence-specific features. We most strongly detected dsRNA for medial regions of the *Tc-zen1* molecule, with a five-fold decrease toward the 3′ and 5′ ends (Figure 1C). We therefore speculated that a dsRNA degradation mechanism may lead to progressive loss of detection from both termini. However, a 5′ terminal amplicon detected stable dsRNA levels throughout embryogenesis (Figure 1B: latter three stages with Fragment 2), arguing for alternative explanations. On further scrutiny, we find that minor differences in amplicon length strongly negatively correlate with amplification efficiency (Figure S2, Supporting Information; ref. [29]). Also, despite primer specificity, we cannot exclude the possibility that our medial amplicon (Fragment 4) may weakly detect the homeobox of the closely related paralogue *Tc-zen2*.[20,51]

Overall, it is striking that long dsRNA is stable in vivo in insect eggs, and our nested amplicon strategy offers new opportunities for dsRNA quantification and long-term tracking. In future, it would be edifying to determine whether maternal transmission also occurs for Dicer-processed siRNA, and the relative contributions of transmitted long dsRNA and siRNAs to gene knockdown in the embryo. For example, whether long dsRNA transmission is sufficient could be evaluated in a Dicer zygotic null background, where only (potentially) transmitted siRNAs are present.

### 3.2. pRNAi Application in Relation to Knockdown Persistence and Female Fecundity

While confirming that pRNAi wanes within individual females (Figure 3; refs. [1, 3, 36]), unexpectedly we find that this only occurs after strong knockdown for nearly 9 weeks—far longer than

was previously shown or assumed. Early research in *Tribolium* reported substantial waning by 3 weeks after injection and complete cessation of knockdown by 5 weeks.[1] Accordingly, developmental genetics research generally examines eggs in the first 4–20 days after injection (e.g., refs. [25, 36, 52]), although ≥90% phenotype penetrance for up to 4.5 weeks has been shown.[11] Differing knockdown durations may reflect differences in injection age (pupal or adult), gene-specific RNAi efficiency,[20,36] and strain-specific rates of waning (Figure 3D). More generally, our results demonstrate the potential for high-efficiency, persistent pRNAi-mediated knockdown, even after a single instance of dsRNA delivery.

It is also surprising that after 50 days there was an abrupt increase in fecundity in both beetle strains used in this study (Figure 3F). It was in this time window of intermediate female age (50–100 days) that we obtained fecundity levels comparable to previous reports, which examined the first 2 months in a third strain (San Bernardino strain: refs. [1, 53]).

These observations highlight within-species variation in the onset and duration of peak fecundity and the rate of RNAi waning. Extrapolation from our study under laboratory conditions (at 30 °C) could also imply longer durations of peak fecundity in natural environments, for slower life cycles at cooler ambient temperatures (e.g., ref. [54]). These factors should be taken into account when planning seasonal management of agricultural pest species by RNAi.[5,6]

### 3.3. Genomic Loss and Ambiguous Homology of SID-1 Emphasizes Its Minimal Relevance for RNAi Outside of Nematodes

The SID-1 channel protein has been part of the standard repertoire of RNAi-associated cellular machinery in surveys of transcriptomes and genomes (e.g., refs. [7, 12, 41, 46]). However, our metazoan-wide appraisal confirms multiple lineage-specific losses of SIL from arthropod genomes (Figures 4 and 5) and that this protein family encompasses homology across SID-1 and TAG-130/CHUP-1 proteins (Figure 5; Figure S1, Supporting Information). This strengthens a cumulative body of evidence in insects for ambiguous homology and limited functional relevance of SIL for RNAi.[4,5,12,42,46]

The loss of SIL proteins is far more pervasive than previously recognized. Among the chelicerates, its absence in the Acari (mites and ticks) contrasts with retention in spiders and scorpions (Figure 4; ref. [47]). Its absence in flies[12,41] may reflect ancestral genomic loss in the wider lineage Antliophora (Diptera, Mecoptera, and Siphonaptera[46]). For other lineages, reports on single or few species noted anecdotal absences, including in the Heteroptera.[7,42,46] A recent review of RNAi specifically in the Hemiptera thus only reported general conservation of SID-1/SIL proteins in this order,[6] without recognizing its wholesale absence in the true bugs (Figures 4 and 5). Species sampling to date also supports SIL loss in the Trichoptera (Figure 4 and ref. [46]: 3 species), which may be further borne out as insect genomic resources continue to grow.

Multiple SIL losses in arthropods may seem surprising compared to its vertebrate-wide retention and the fact that nematodes and arthropods are more closely related as fellow Ecdysozoa

(Figure 4). This could suggest a higher rate of evolutionary divergence in arthropods against a backdrop of bilaterian-wide conservation. In fact, vertebrate protein homology suffers from the same ambiguities as analyses with arthropod proteins (Figure S1, Supporting Information). Vertebrate Sidt proteins show greater sequence similarity in certain functional motifs with TAG-130/CHUP-1 proteins, recognized for their role in cholesterol uptake.[55] Furthermore, recent cell culture work suggests that prior evidence for dsRNA uptake by Sidt/CHUP-1 may have detected a secondary consequence of dsRNA association with imported cholesterol,[56] calling Sidt molecular function into question. Overall, this is conceptually similar to the macroevolutionary "functional lability" and repeated lineage-specific loss of RNA-dependent RNA polymerases (RdRPs),[57] another component of systemic RNAi in some species (see below).

In *C. elegans*, SID-1 is required for the systemic spread of RNAi within somatic tissues and the pRNAi effect in offspring.[41] Yet, despite the absence of any SID-1/SIL protein, the Heteroptera are highly sensitive to RNAi (reviewed in ref. [58]). Knockdown is effective and systemic within the bodies of individual heteropteran nymphs.[59] pRNAi can achieve complete phenotypic knockdown in >95% of progeny for at least 3 weeks.[60]

Thus, just like other nematode SID proteins,[4,5,43] SID-1 should be retired from general inclusion among the insect RNAi repertoire.

### 3.4. The Power of Orthology Clustering, in Context

As discussed, some of our key insights into the taxonomic distribution of SID-1 were already documented on an anecdotal level in a range of published studies, but they had not been integrated. We show that metazoan-wide orthology clustering[45] combined with taxonomically informed visualization (Figure 4) can reveal previously unappreciated macroevolutionary patterns of protein origin, conservation, duplication, and loss across disparate lineages such as insects and vertebrates. With corroboration from additional lines of evidence including protein member curation, genome searches, phylogenetics, and literature surveys (Figure 5), this is a powerful approach.

Such rapid phylogenomic profiling (Figure 4) could be widely applied to whole suites of proteins, providing criteria for candidate gene selection alongside standard gene ontology (GO) features such as molecular function (transmembrane receptor) or biological process (receptor-mediated endocytosis). And, while our focus is the insects in general, visualization can be customized for other taxa of interest (e.g., Vertebrata, Hymenoptera), particularly as the number and diversity of sequenced genomes increases.

Orthology clustering across distantly related species requires care. Whereas wholesale loss or duplication in a clade is convincing, taxonomically scattered copy number changes may reflect genuine evolutionary change in undersampled lineages or limitations in individual species' data quality. Manual curation is necessary to eliminate redundant isoforms, which inflate copy number (Figure 5A), and incomplete or suspiciously large and divergent proteins, which often reflect inaccurate gene model annotation[48,49] and can skew phylogenetic analysis (see Experimental Section). Second, each taxonomic level of orthology clustering is an independent analysis. At wider taxonomic levels, groups of single-copy orthologs often gain divergent within-species homologs and appear multi-copy due to greater sequence divergence between homologs in distantly related species. The inclusion of divergent vertebrate Vldlr proteins within the metazoan-level orthology group for VgR exemplifies this (Figure 4). The challenge of reconciling clustering analyses across taxonomic levels is a known, but perhaps not widely appreciated, issue.[61] Clarification of orthology is possible by prioritizing taxonomically restricted clustering results and then progressively adding wider taxa (e.g., from Insecta to Metazoa, Figure 4), supported by phylogenetic analysis (Figure 5). However, the SID-1 and TAG-130/CHUP-1 proteins are particularly recalcitrant, forming a single orthology group even within the Nematoda alone.

### 3.5. How Can pRNAi Persistence Be Reconciled with dsRNA Cellular Processing and Maternal Transmission?

Our unexpected finding that the long dsRNA molecule is maternally transmitted into eggs, contributing to depletion of maternal dsRNA levels, is difficult to reconcile with pRNAi persistence for months (Figures 1–3). We also find limitations in attributing dsRNA cellular transmission to specific import proteins (Figures 4 and 5). Furthermore, biochemical, physiological, and cellular studies on dsRNA processing highlight where dsRNA is *not* located, rather than how it is delivered to Dicer to trigger RNAi. To conclude, we discuss how our observations fit into the wider framework of outstanding major questions on systemic parental RNAi in insects (**Figure 7**).

Upon injection into the female's body cavity (Figure 7A), dsRNA spreads throughout the circulatory system. However, it rapidly clears—on the scale of minutes to hours—from the hemolymph due to cellular uptake and degradation (Figure 7B; refs. [21, 23, 62]). In *Tribolium*, substantial activity of endogenous dsRNases is documented in the gut and implicated in the hemolymph.[63] Also, the ovary represents just one organ in the female body in which dsRNA uptake occurs. In effect, the germline competes with other cell types for dsRNA. Particularly when it is distal to the site of dsRNA injection, it may be less sensitive or even refractory to RNAi.[9,23] Injection of dsRNA for pRNAi is highly effective in practice, but not without limitations.

Second, the dsRNA received by the insect ovary represents a non-renewable resource. In this and other studies, pRNAi is achieved after a single injection, providing a finite number of dsRNA molecules. That starting pool is amplified by RdRPs in plants, nematodes, and possibly fungi.[57,64–66] This property can be exploited in planta for sustained delivery of non-endogenous transcripts in RNAi-based pest control.[64] However, there is no evidence to date for dsRNA amplification in insects (reviewed in refs. [57, 63]). Also, amplification in other species generally or exclusively involves siRNA synthesis,[64–66] which contrasts with our detection of ≥100-bp RT-qPCR amplicons spanning full-length long dsRNAs (Figures 1 and 2).

Next, there are uncertainties as to how cellular uptake of long dsRNA is accomplished (Figure 7C). In principle dsRNA could be shuttled into the oocyte after uptake by the nurse cells or the follicular epithelium, or it could be directly imported by the

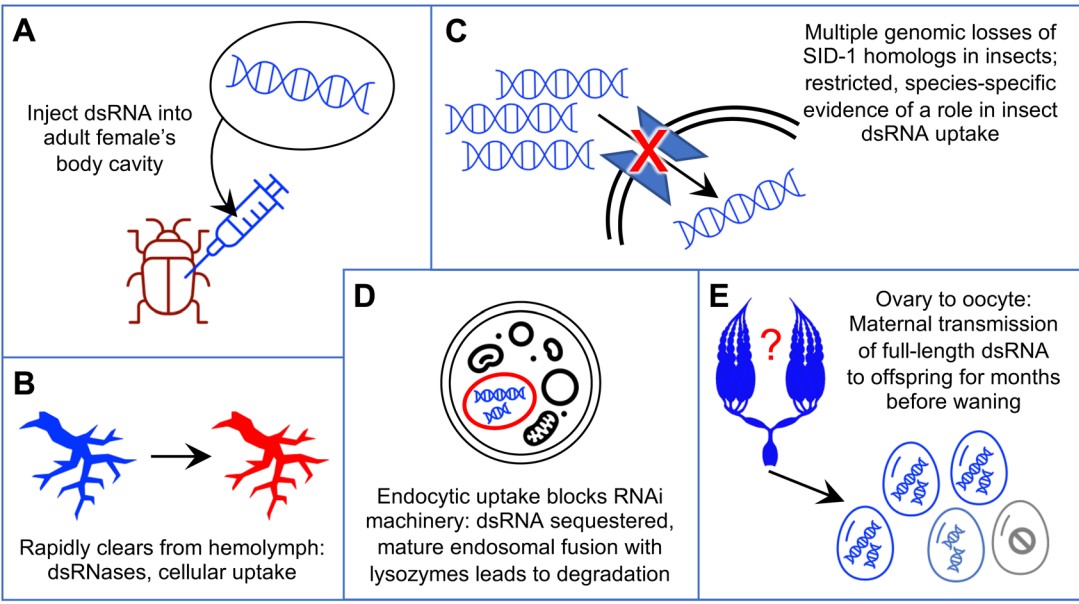

**Figure 7.** Unresolved features of systemic parental RNAi. Where is the dsRNA stored long-term in the mother without degradation and with continuous transmission to eggs? Cartoons represent the progression of dsRNA from initial injection (A), through the mother's tissues (B) and cells (C,D), to the oocytes (E). In detail, we depict injection into the abdominal body cavity (A); clearance from the hemolymph (B, schematized representation of the dorsal vessel (heart) and major circulatory system branching structures such as arteries[91]); uptake across the plasma membrane into individual cells, which may be receptor-mediated (C); and potential sequestration from the cytosol in endosomes (D). Despite challenges associated with each of these steps (see Section 3.5), systemic parental RNAi involves dsRNA transmitted from the ovary into oocytes for months (E). Presence of dsRNA is represented in blue, with specific cell- and tissue-scale challenges to its transmission shown in red, and with final waning of pRNAi indicated by pale blue and grey. Clip art images reproduced and modified from Microsoft PowerPoint 2021, v. 16.52; ovary silhouette based on image at https://cronodon.com/BioTech/Insect_Reproduction.html, adapted with permission.

oocyte during patency, when intercellular openings in the follicular epithelium confer direct access to the hemolymph. However, neither SID-1 for cellular uptake (discussed above) nor VgR for oocyte endocytosis seems to be the effector. In *C. elegans*, co-accumulation of dsRNA and vitellogenin in oocytes suggested a common import mechanism for these molecules.[22] However, the VgR receptor is hexapod-specific (Figures 4 and 5), arguing against a conserved mechanism associated with invertebrate vitellogenin transport. Furthermore, trials with labeled dsRNA demonstrated its exclusion from oocytes during vitellogenesis.[23] On the other hand, SID-1 and VgR are two candidates among many potential receptor proteins. Clathrin-dependent endocytosis is required for within-individual larval RNAi in *Tribolium*,[24] and such mechanisms may also be applicable for pRNAi.

More generally, endocytosis has long been recognized as a potential mechanism for dsRNA uptake, but it has its own cellular challenges (Figure 7D; reviewed in refs. [4, 5, 42]). First, if dsRNA is sequestered within an endosome, it is inaccessible for processing by Dicer in the cytosol, and the mechanism of selective endosomal escape of dsRNA is unknown. Species-specific levels of dsRNA sequestration have been correlated with susceptibility to RNAi.[5] Second, endosome maturation culminates in fusion with a lysosome, targeting all contents for degradation.[4] Thus, endosomes do not seem suitable as long-term, slow-release reservoirs for pRNAi. Beetles including *Tribolium* appear to have low levels of endosomal sequestration, but those studies were performed in larvae (reviewed in ref. [5]). Further investigation of maternal reproductive tissues may reveal alternative, germline-specific mechanisms of dsRNA retention and cell-to-cell transmission. This would be fully consistent with the growing body of evidence for the tissue-specific as well as stage-specific nature of RNAi (e.g., discussed in refs. [9, 23, 67]).

Finally, dsRNA's journey from maternal injection through successful embryonic knockdown requires two levels of cellular transmission (Figure 7E). After dsRNA is delivered into the oocyte, cellular uptake must happen again: when dsRNA within the yolky oocyte is taken up by the embryonic cells, where knockdown is finally achieved. As maternal injection can lead to deposition of labeled oligonucleotides in the yolk without embryonic uptake,[68] this step also cannot be taken for granted. In summary, while we continue to successfully use pRNAi for developmental genetics research and in devising new and improved strategies for pest management, there remain many aspects of dsRNA transport and systemic propagation that await explanation.

## 4. Experimental Section

*Tribolium castaneum (Herbst) Stocks and Genomic Resources*:    All beetle stocks were kept under standard culturing conditions[13] at 30 °C, 50 ± 10% RH. The lines used for the RT-qPCR assays were San Bernardino (SB) wild type[13] and nuclear GFP (nGFP).[33] For the RNAi penetrance time course experiments, Strain 1 was a heterozygous cross of the enhancer trap lines G04609 (females)[35] and HC079 (males)[30], both in the *pearl* white-eyed mutant background.[69] Strain 2 was the LifeAct-GFP line, in a rescued *vermillion white* background.[70]

Sequence data for the target genes in this study were based on the latest genome assembly and official gene set (OGS3)[71]:

*Tc-zen1* (TC000921)[20,26], *Tc-chitin synthase 1* (*Tc-chs1*, TC014634)[27], *Tc-Ribosomal protein S3* (*Tc-RpS3*, TC008261)[25], *Tc-germ cell-less* (*Tc-gcl*, TC001571)[39], and *Tc-tailup* (*Tc-tup*, TC033536).[15,37] Details of primers and amplicon sizes are presented in Table S1, Supporting Information, also for the transgene *DsRed2* (based on the *piggyBac* mutator construct: GenBank accession EU257621.1).

*RT-qPCR Experiments*: Embryos were collected over a period of 20 days after injection. Knockdown efficiency was ensured by: manual assessment of serosal cuticle structure (eggshell rigidity) for *Tc-zen1*[11,20] and *Tc-chs1*,[27] detection of fluorescent signal for *dsRed*,[34,35] and by RT-qPCR for all genes. To evaluate *DsRed* knockdown efficiency by fluorescence screening, only larvae were scored to ensure all offspring had successfully completed embryogenesis and were thus old enough to produce strong 3xP3-DsRed signal.

RT-qPCR and data analysis were performed as described, including TRIzol extraction, DNase treatment, gDNA quality control checks, cDNA synthesis, and Fast SYBR Green detection on an Applied Biosystems 7500 Fast cycler (reagents: ThermoFisher Scientific; TURBO DNAfree Kit, Applied Biosystems; SuperScript VILO cDNA Synthesis Kit, Invitrogen; Life Technologies; respectively).[20,25] All samples were run in triplicates (technical replicates) with three samples per treatment (biological replicates). *Tc-RpS3* was used as the reference gene, this being established as more stable across embryogenesis as a single reference gene compared to several alternatives with pairs of reference genes or seven other single genes.[25] Raw data were analyzed using LinRegPCR v12.16[72,73] and the expression ratio (R) was calculated using the $\Delta\Delta$Ct method, according to the equation:

$$R = \frac{\left(E_{\text{target}}\right)^{\Delta CP_{\text{target}}(\text{control}-\text{sample})}}{\left(E_{\text{ref}}\right)^{\Delta CP_{\text{ref}}(\text{control}-\text{sample})}} \tag{1}$$

where $E$ is the mean efficiency of the corresponding amplicon as calculated by LinReg and $CP$ is the mean CP of the three technical replicates (after passing quality control in LinReg). The control sample was a pool of all samples (WT and RNAi; all time points; all biological replicates) of the respective experiment (i.e., RNAi knockdown of a given gene: *Tc-zen1*, *Tc-chs1*, or *dsRed*). The % of wild type was calculated by dividing $R_{RNAi}$ by $R_{WT}$ for the same time point and sample collection date, where both $R$ values were relative to the control sample.

*Parental RNAi*: Parental RNAi was performed as in previous work.[25] In detail, RNA purification and cDNA synthesis were performed as described above for the RT-qPCR experiments (TRIzol extraction, Invitrogen SuperScript VILO cDNA Synthesis Kit). Linear template for dsRNA synthesis was generated by PCR amplification (Sigma REDTaq ReadyMix PCR Reaction Mix, #R2523) of embryonic cDNA, using universal primers to add T7 RNA polymerase docking sites (Table S1, Supporting Information). The dsRNA was synthesized with the Ambion MEGAscript T7 Transcription Kit (#AM1334, supplied from Life Technologies or ThermoFisher). dsRNA was resuspended in $H_2O$ and injected at a concentration of $\approx 1$ µg µL$^{-1}$ (range: 900–1100 ng µL$^{-1}$). Volume injected per female was $\approx 400$ nL.

Beetles were sexed as pupae (distinguished by genital morphology) and allowed to mature to adulthood. Females were anesthetized on ice and dsRNA was injected into the abdomen. Uninjected females served as wild type controls. Gene-specific knockdown phenotypes were confirmed based on published resources for all genes, using the specific assays described for each of the RT-qPCR and time course experiments. As *Tc-tup* has thus far only been characterized in a high throughput screening analysis,[15,37] two non-overlapping fragments (NOFs) of dsRNA were used in the experiments (NOF1 for Experiments 1 and 2, NOF2 for Experiment 3: see Table S1, Supporting Information). No quantitative or qualitative phenotypic differences were found between the non-overlapping fragments.

*RNAi Penetrance Time Course Experiments*: Larval cuticle preparations were used to monitor phenotype penetrance over time after a single injection of dsRNA into the adult female. A cuticle assay is highly effec-

tive even with limited embryonic material, which was important in the months-long experiments because female survival and fecundity decline over time.[74] Moreover, *Tc-tup* and *Tc-gcl* provide clear cuticle readouts, whereas RNAi for each of the RT-qPCR target genes can result in non-lethal knockdown that must be analyzed at specific developmental stages (Figure 2C–F; ref. [27]).

Eggs were collected at regular intervals and maintained under standard culturing conditions until a minimum age of ≥4 days after egg lay, to ensure time for larvae to hatch. Larval cuticles were then prepared as described previously.[15] Briefly, eggs and larvae were dechorionated in bleach (VWR #L14709.0F, sodium hypochlorite (11–14% Cl$_2$) in aqueous solution), rinsed in tap water, and mounted on slides in 1:1 lactic acid:Hoyer's solution.[75] Slides were cured overnight at 60 °C to fully clear soft tissues. Slides were then scored under incidental white light on stereomicroscopes, distinguishing six categories: wild type larvae, unhatched wild type (post dorsal closure with no apparent defects, but still at least partially within the vitelline membrane), gene-specific phenotype category 1 (generally a larger body size), gene-specific phenotype category 2 (generally a smaller and less well formed body), non-specific defects, or no larval cuticular material ("empty egg," indicative of unfertilized eggs or early embryonic lethality). Statistics on penetrance compare wild type with gene-specific knockdown, combining each of the first two pairs of categories while for simplicity omitting the latter two, minor categories. The time point of a sample represents the start of the egg collection period (e.g., data at 3 days post injection (dpi) represent the sample collected 3–4 dpi in Experiment 1, Figure 3D). Egg collection intervals were extended, or consecutive collections were pooled, to ensure sample sizes of ≥10 offspring per treatment condition for each time point.

Experiments were conducted until three egg collections contained only hatched larvae and the knockdown effect was deemed to have fully waned. Throughout the experiments, dead adult beetles were periodically removed and sexed to note female-specific lethality (males have a darkened cuticular sex patch on the inner/proximal side of the first leg pair; this was absent in females: https://www.ars.usda.gov/plains-area/mhk/cgahr/spieru/docs/tribolium-stock-maintenance/#sexing [last accessed October 15, 2021]).

To assay females of different ages, adult beetles were maintained continuously under standard culturing conditions at 30 °C until injection. Female age was calculated from the last date when beetles in the experimental cohort were sexed as pupae, reflecting a minor overestimation (≤5 days) relative to eclosion of the adult for some individuals in the cohort. The females used in Experiments 1 and 2 derive from the same cohort and were sexed at the same time.

*Microscopy*: Images were acquired on an epifluorescent microscope with structured illumination (Zeiss Axio Imager.Z2 with Apotome.2). Red fluorescence signal in the eyes and ventral nerve cord was used to evaluate *DsRed* RNAi, with green fluorescence from the ubiquitous nGFP signal in this transgenic line serving as an internal control. Representative cuticle images were acquired with GFP acquisition settings to detect cuticle autofluorescence, presented as maximum intensity projections from the acquired z-stacks.

*Orthology Distribution, BLAST, and Phylogenetic Evaluations*: Orthology groups were examined in OrthoDB v. 10.1,[45] comparing the independent orthology clustering analyses at taxonomic levels including Metazoa, Arthropoda, Hexapoda, Insecta, Hemiptera, Coleoptera, Nematoda, and Vertebrata. Minor changes in species membership, copy number, and protein ID were noted between the independent orthology clustering analyses conducted at the various taxonomic levels, which is a known issue for orthology clustering (discussed in ref. [61]). In all cases, data at the most taxonomically restrictive level (last common ancestor, LCA, level) were used as the most specific and reliable. For the genes examined here (Figure 4), orthology clustering was very robust, with only minor differences (e.g., Figure 4: asterisk and legend note for VgR).

Curation of protein sequences obtained from orthology groups involved visual inspection of the protein size and sequence in order to remove partial and redundant isoforms. In choosing appropriate protein members of an orthology group for use in phylogenetic analyses, visual inspection of

www.advancedsciencenews.com

ADVANCED
GENETICS

www.advgenet.com

multiple sequence alignments and preliminary trees were used to identify and cull divergent (long branch) proteins and overly large proteins (which may reflect erroneous protein fusion or other model annotation errors such as inclusion of extraneous predicted exons).

Protein sequences were aligned for manual inspection in ClustalW,[76] at https://www.genome.jp/tools-bin/clustalw [last accessed October 15 2021]. Phylogenies were generated at Phylogeny.fr with default settings (alignment with MUSCLE 3.8.31, phylogeny with PhyML 3.1/3.0 aLRT, and tree rendering with TreeDyn 198.3).[77]

Genome assemblies were examined by BLAST, supported by visual inspection of hits with respect to the assembly, gene model predictions, and expression evidence tracks in the Apollo genome browsers, hosted at the i5K@NAL workspace.[78] Species sampling involved a particular focus on the Heteroptera[48,79–82] and selected species from other orders (Thysanoptera,[83] Hymenoptera,[84] Coleoptera[85,86]). The genome assembly versions interrogated by tBLASTn are detailed in Table S2, Supporting Information.

*Statistical Analysis*: Sample sizes and numerical handling are detailed above in the sections on "RT-qPCR experiments" and "RNAi penetrance time course experiments," as well as in the associated figure legends (Figures 1–3). Briefly, RT-qPCR was conducted with triplicate technical and biological replicates, with outliers evaluated with LinRegPCR v12.16.[72,73] Figures present mean expression levels, and error bars indicate ± one standard deviation. Phenotypic scoring of loss of *DsRed* by fluorescence microscopy was based on sample sizes of $n > 150$ per treatment condition. Larval cuticle phenotypic scoring used sample sizes of $\geq 10$ offspring per treatment condition for each time point.

## Supporting Information

Supporting Information is available from the Wiley Online Library or from the author.

## Acknowledgements

The authors thank Stefan Koelzer for generating and processing RNAi samples for RT-qPCR; Siegfried Roth for diverse discussions on insect pRNAi; Robert M. Waterhouse for assistance with and discussions on OrthoDB; Robert M. Waterhouse, Ruixun Wang, and two anonymous reviewers for detailed and helpful scrutiny of the manuscript; and the Stancliffe Institute for Remote Working for infrastructural support. The authors also thank Sebastien Santini (CNRS/AMU IGS UMR7256) and the PACA Bioinfo platform (supported by IBISA) for the availability and management of the phylogeny.fr website used for the phylogenetic analyses. Finally, the authors thank Dominik Stappert for originally designing the long template primer pair for *Tc-zen1* during his doctoral work,[74] as this has proven extraordinarily fruitful over the years. The research was funded by the German Research Foundation (Deutsche Forschungsgemeinschaft), through Emmy Noether Program grant PA 2044/1-1; the University of Warwick, through a Warwick Institutional Research Support Fund award; and the Biotechnology and Biological Sciences Research Council (BBSRC UKRI), through grant BB/V002392/1, to K.A.P.

## Conflict of Interest

The authors declare no conflict of interest.

## Author Contributions

T.H. and K.A.P. contributed to the conceptualization and primary writing. T.H., K.D.N., and K.A.P. conducted experiments and analyzed data. T.H., K.D.N., and K.A.P. contributed to the discussion, review, and editing of the manuscript.

## Data Availability Statement

The data that support the findings of this study are available in the article and in the cited public repositories, supported by repository versioning documentation in the Experimental Section and Supporting Information.

## Peer Review

The peer review history for this article is available in the Supporting Information for this article.

## Keywords

dsRNA uptake, insects, parental RNAi, phylogenomic profiling, RT-qPCR, systemic RNAi, *Tribolium castaneum*

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
