## [**Supplementary Information**: Record of Transparent Peer Review · Advanced Genetics]

Record of Transparent Peer Review

Persistent parental RNAi in the beetle *Tribolium castaneum* involves maternal transmission of long double-stranded RNA

Thorsten Horn, Kalin D. Narov, and Kristen A. Panfilio*

*Corresponding

Review timeline:	Date Submitted: 12-Nov-2021
	Editorial Decision: 14-Dec-2021
	Revision Received: 31-Jan-2022
	Accepted: 01-Feb-2022

Editor: Myles Axton

1 st Peer Review	15-Nov to 13-Dec-2021
-----------------------

Reviewer #1

Understanding the molecular mechanisms underlying parental RNAi in insects is fundamental to applying it in basic research and pest control in the field. The observation that different probes provide different results in RT-PCR assays is not only interesting but has significant impact to the design and interpretation of experimental results.

1.1 Therefore is it vital to provide clear descriptions of the results that are not conflated with the authors favored interpretation of said results.

1.2 A general note, this manuscript reads like three separate studies: dsRNA in embryos, endurance of pRNAi effects, and computational phylogenetic of Sid and VgR candidate genes that have been considered in the past, but mostly ruled out. Integrating these stories would strengthen this presentation.

1.3 For example, the first description of the observed phenomenon (shown in Figure 1) is a little confusing.

If Tc-zen1 is highly expressed early (8-24 hr) and later undetectable or expressed at very low levels (Line 155), there cannot be "strong RNAi knockdown at all stages" as stated in line 131. Later, this description is corrected as "and no later overexpression", but even that seems to be an interpretation. Isn't the observed result simply higher levels of RNA in RNAi individuals than in WT?

It would be clearer to remove the first phrase (a misleading interpretation of the results and simply leave the description that follows in the next sentence.

1.4 Line 144 "ostensible overexpression" is an interpretation of the result, which is simply higher levels of RNA in RNAi embryos throughout embryogenesis. Is this a strawman argument? An interpretation to disprove? Seems unnecessary. What is meant by unmasked??? As the authors note, higher levels of RNA are also detected in RNAi embryos at the earliest stage, when using the completely nested probes compared to partially nested probes. However, it appears to have been misinterpreted in previous studies.

Line 153 "certain amount of dsRNA transmitted, that may be masked by high endogenous expression" again, It isn't masked, so much misinterpreted and now revealed by choice of assay fragment.

1.5 What is the amount of dsRNA transmitted to the embryo? Is the amount injected into adult females saturating the system? A few dosage experiments would help clarify this. Do pRNAi effects wane more quickly with lower dose, or is RNAi effect weaker (in phenotype and RNA levels) in beginning?

1.6 line 175. If wild type expression is negligible at 16-24 hours, how can there be efficient knockdown, do you see higher levels of mRNA to be knocked down? The results seem to be interpreted rather than described and then interpreted. The data reveal

increased levels of RNA in RNAi individuals compared to wt, and only with probes internal to the dsRNA. This could be interpreted as coming from residual dsRNA. Similar results with multiple probes spanning the dsRNA suggest the residual RNA is likely full length.

1.7 Line 201 "overexpression" is an interpretation, while the observed result is higher levels of RNA in RNAi individuals.

1.8 Line 406 "primer design may be constrained" is rather vague. It is hard to tell if this is a recommendation, observation of past experiments or problem that might be encountered. Please specify or clarify.

1.9 Line 552 "Our unexpected finding that the long dsRNA molecule is maternally transmitted into eggs, thereby depleting maternal dsRNA levels, is difficult to reconcile with pRNAi persistence for months (Figs. 1-3)". This statement is difficult to accept. First they did not directly demonstrate that the maternal levels of dsRNA do actually decline as the pRNAi effect wanes. Second, what are the alternatives? If the dsRNA is processed prior to being imported into the embryos, wouldn't that also deplete the maternal levels of long dsRNA?

1.10 Figure 6 is not referred to in the text. Relying on the figure legend for context, it appears to be showing the amounts of endogenous mRNA available to be knocked down at different time points. It is not clear how much dsRNA is present at these sample timepoints and how detection of it would affect the results of the assay. In particular, isn't there dsRNA present at the narrow sampling timepoint that obscures the true knockdown levels? And if the dsRNA is knocked down how can it "appear" at the longer sample timepoint? It would be interesting to see what the authors think is the relative amount of dsRNA present at these sample time points. Also please incorporate this figure and the data within in the main text.

The phylogenetic analysis is very thorough and convincing, although it only considers previously suggested candidate genes have largely been removed from consideration as candidate genes involved in the transmission of the dsRNA into the embryos.

Reviewer #2

Review of "Persistent parental RNAi in the beetle *Tribolium castaneum* involves maternal transmission of long double-stranded RNA" by Thorsten Horn, Kalin D. Narov and Kristen A. Panfilio for publication in *Advanced Genetics*.

The authors performed a trilateral investigation of the molecular mechanism of pRNAi in the model insect *Tribolium castaneum* that combines experimental results with comparative genomics. They present a RT-qPCR strategy that unmask the presence of dsRNA distinct from endogenous mRNA. This method revealed a strong pRNAi that can persist for months. In addition, the authors present phylogenomic profiling of possible receptor proteins for cellular uptake of long dsRNA into the egg.

The manuscript provides important basic knowledge on molecular mechanisms of dsRNA transmission during systemic pRNAi. The paper is very clear and nicely written, I only some minor comments, and suggestions (see below).

2.1 If the manuscript is not already on the maximum word count, I would suggest including more information in the 'Parental RNAi' section of the M & M (e.g. injected amount of dsRNA, transcription of dsRNA etc.). The authors refer to citation no 25, which is than referring to the manuscript of van der Zee et al. (2005). Referring from one publication to another is a bit annoying, including all the needed information in just one publication, if possible, is much more convenient for the reader.

2.2 Is Line 135: 'semi-nested amplicon' - is this fragment 1?

2.3 In line 136&137 the authors refer to an 'original assay'. Could the authors include a specific reference? I was not sure which study is meant, ref no 20? Ref no 25?

2.4 The authors might consider including studies performed by the group of Palli about endosomal entrapment in the discussion e.g.: Yoon JS, Gurusamy D, Palli SR. (2017) doi: 10.1016/j.ibmb.2017.09.011

2.5 Figure 6 is missing the meaning of the abbreviation 'KD' in the figure legend.

2.6 Figure 7 looks a bit unorganized and to me it is not clear what is displayed in 7B.

1st Editorial Decision

14-Dec-2021

Revise according to the two reviewers' comments and editorial advice below

Editor's understanding of the reviews

Reviewer #1 Recommends: Major Revision

Reviewer #2 Recommends: Minor Revision

Author's Response to 1st Review

31-Jan-2022

We thank the reviewers and editor for their careful reading of our manuscript and constructive questions and suggestions. Our responses are detailed below in blue text, interleaved with the reviewers' original reports.

ggn2.202100064 AU: Panfilio, Kristen		Advanced Genetics Editor: Myles Axton	
Title: Persistent parental RNAi in the beetle Tribolium castaneum involves maternal transmission of long double-stranded RNA These are the main reviewer recommendations that the editors believe will make the biggest improvement to this article. Please do address all reviewer comments listed in the decision letter in your point-by-point response (you may continue this table to do so if you wish). We hope this summary helps you to understand our decision and expedites the revision process. We value feedback from author and referees alike. AdvGenet@wiley.com			
Editorial decision: Revise according to the two reviewers' comments and editorial advice below Editor's understanding of the reviews Reviewer #1 Recommends: Major Revision Reviewer #2 Recommends: Minor Revision			
Reviewer comments	Editor recommendation	Author reply	Changes to Manuscript
1.5 What is the amount of dsRNA transmitted to the embryo? Is the amount injected into adult females saturating the system? A few dosage experiments would help clarify this. Do pRNAi effects wane more quickly with lower dose, or is RNAi effect weaker (in phenotype and RNA levels) in beginning? 1.9 First they did not directly demonstrate that the maternal levels of dsRNA do actually decline as the pRNAi effect wanes. Second, what are the alternatives? If the dsRNA is processed prior to being imported into the embryos, wouldn't that also deplete the maternal levels of long dsRNA?	ED1 a) discuss how much dsRNA is transmitted relative to the injected dose. How does this affect interpretation of pRNAi vs RNAi effect? b) Is it possible to show by quantitation that the dsRNA decline is due to pRNAi effect or processing?		
1.1 is it vital to provide clear descriptions of the results that are not conflated with the authors favored interpretation of said results. 1.3 Isn't the observed result simply higher levels of RNA in RNAi individuals than in WT? 1.4 What is meant by unmasked? 1.6 The results seem to be interpreted rather than described and then interpreted 1.7 "overexpression" is an interpretation, while the observed result is higher levels of RNA in RNAi individuals.	ED2 Report the results in terms of higher and lower levels of RNA species detected in each case. Reserve interpretation for the end of the Results (after the figures have been introduced) or in the Discussion. You can make clear which RNA species are identified by your new probes that were missed in prior literature. Avoid "unmasked" unless you mean RNA that is functionally inaccessible because of secondary structure or protein complexes. Avoid "reveal" in favor of "show". Introduce inferences as such rather than using data to "uncover" your prior hypotheses.	See below the point-by-point responses, using the editor's numbering system.	Detailed below, with page numbers referring to the revised manuscript Word document with changes highlighted in yellow.
2.1 I would suggest including more information in the 'Parental RNAi' section of the M & M (e.g. injected amount of dsRNA, transcription of dsRNA etc.). ..., including all the needed information in just one publication, ... 2.3 In line 136&137 the authors refer to an 'original assay'. Could the authors include a specific reference?	ED3 There is no formal word count and references are unlimited. We would prefer you cite published methods with enough detail that the experiments can be understood without reading the original methods that may be in several publications.		
2.4 The authors might consider including studies performed by the group of Palli about endosomal entrapment in the discussion	ED4 the editor leaves it up to the author to choose relevant references, but the reviewer is suggesting a possibly relevant mechanism for compartmentation or protection of specific RNA forms.		

Reviewer #1

Understanding the molecular mechanisms underlying parental RNAi in insects is fundamental to applying it in basic research and pest control in the field. The observation that different probes provide different results in RT-PCR assays is not only interesting but has significant impact to the design and interpretation of experimental results.

1.1 Therefore is it vital to provide clear descriptions of the results that are not conflated with the authors favored interpretation of said results.

A1.1) We agree with the reviewer that factual reporting of results and subsequent interpretation should be clearly distinguished. As the reviewer surmises, our challenge is to address what “appears to have been misinterpreted in previous studies” (comment 1.4, below). The shorthand phrasing we had deliberately used proved effective in communicating this study to scientific peers, but we accept that this may not generalize well beyond the subdiscipline of developmental genetics.

Our use of the term “overexpression” reflected first-order up/down interpretation of the ratio (R_{RNAi}/R_{WT}), using standard terminology in developmental genetics. No “favored interpretation” of the authors was implied. To reconcile these considerations, we now include a brief paragraph explaining ratio interpretation at the beginning of the first results section (page 4, second paragraph), and we nonetheless (a) remove all subsequent use of the term “overexpression” and similar language in the results and figure legends, and (b) first present results (“We observed...”, p. 4, last paragraph) before offering local interpretation in a separate paragraph at the end of the section (“Thus, we infer...”, p. 5, second paragraph). In the Discussion, we do retain the phrase, “or even to erroneous interpretations of target gene overexpression” in expounding on the issue.

Similarly, “unmasking” of dsRNA presence in eggs is relative to standard interpretations in the literature. For clarity, we have replaced all instances of this word with phrases such as “distinguish the presence of double-stranded RNA (dsRNA) from endogenous mRNA” (abstract, p. 1) and “[RNA] levels that may be overlooked” (results, p. 5, second paragraph), or added complete new sentences instead (end of Figure 6 legend).

1.2 A general note, this manuscript reads like three separate studies: dsRNA in embryos, endurance of pRNAi effects, and computational phylogenetic of Sid and VgR candidate genes that have been considered in the past, but mostly ruled out. Integrating these stories would strengthen this presentation.

A1.2) We acknowledge that this study is tripartite (first line of discussion), but throughout we make the scope of the work explicit (abstract, last paragraph of introduction, first paragraph of discussion). Furthermore, we take care to explain our intellectual motivation and the narrative progression for each part at the transition points in the results (first paragraphs of sections 2.4 and 2.6). All of three parts are integrated in the final summary Figure 7.

1.3 For example, the first description of the observed phenomenon (shown in Figure 1) is a little confusing. If Tc-zen1 is highly expressed early (8-24 hr) and later undetectable or expressed at very low levels (Line 155 [NB line 145?]), there cannot be “strong RNAi knockdown at all stages” as stated in line 131. Later, this description is corrected as “and no later overexpression”, but even that seems to be an interpretation. Isn't the observed result simply higher levels of RNA in RNAi individuals than in WT? It would be clearer to remove the first phrase (a misleading interpretation of the results and simply leave the description that follows in the next sentence.

A1.3) We thank the reviewer for flagging our conflation of phenotypic evidence for RNAi knockdown with RNA levels at different developmental stages. (Indeed, in Discussion section 3.1 and associated Figure 6 we discuss stage-specific implications for RT-qPCR experimental design: see below, response A1.10.) This has been rewritten as, “we obtain lower RNA levels in RNAi samples than WT samples at all assayed stages”,

before only referring to knockdown (% WT) for the 8-24 h sample (p. 4). Please see our response A1.1 on the prior use of the term “overexpression”.

1.4 Line 144 "ostensible overexpression" is an interpretation of the result, which is simply higher levels of RNA in RNAi embryos throughout embryogenesis. Is this a strawman argument? An interpretation to disprove? Seems unnecessary. What is meant by unmasked??? As the authors note, higher levels of RNA are also detected in RNAi embryos at the earliest stage, when using the completely nested probes compared to partially nested probes. However, it appears to have been misinterpreted in previous studies. Line 153 "certain amount of dsRNA transmitted, that may be masked by high endogenous expression" again, It isn't masked, so much misinterpreted and now revealed by choice of assay fragment.

A1.4) Please see our response A1.1 on the prior use of the terms “overexpression” and “unmasked”. We replaced the sentence, “This implies that the ostensible overexpression represents the unmasked detection of dsRNA specifically at older developmental stages when wild type expression is low”, with the new text, “At older developmental stages when wild type expression is low or undetectable, the dsRNA would constitute the majority or all detected RNA. This is consistent with the observed higher levels of RNA in RNAi than WT samples (Figure 1B: yellow vs. red plot lines, developmental time $\geq 16-24$ h)” (p. 5, second paragraph).

1.5 What is the amount of dsRNA transmitted to the embryo? Is the amount injected into adult females saturating the system? A few dosage experiments would help clarify this. Do pRNAi effects wane more quickly with lower dose, or is RNAi effect weaker (in phenotype and RNA levels) in beginning?

A1.5) This is an intriguing and wide-ranging set of questions. We do think that the system is being saturated, but quantifying (a) the exact amount of dsRNA transmitted to the ovary and then to the eggs, and (b) how this class of molecule directly contributes to gene knockdown would require development and optimization of methods beyond those used in the current study.

In terms of saturation, we had already noted “transmitted dsRNA stably persists in the egg throughout this interval (Figure 1B: yellow plot line, $\geq 16-24$ h)”. Immediately following this observation, we now add the explicit comment, “Given phenotypic and molecular evidence (at 8-24 h) of RNAi knockdown, RNA degradation occurs in the embryo. Thus, our observation of stable dsRNA levels throughout embryogenesis suggests that dsRNA is transmitted into eggs at saturating levels that exceed our ability to detect a drop in dsRNA levels over time” (p. 5, second paragraph).

New methods would be necessary for objective quantification of dsRNA transmitted to the egg compared to our injection of approximately 400 ng dsRNA into the mother’s body cavity. Our RT-qPCR strategy of comparing nested and semi-nested amplicons is necessarily a relative measure, and these samples’ values are already expression ratios relative to the pooled control per qPCR plate, and where all samples are normalized relative to the reference gene. In light of dsRNA distribution throughout the mother’s body, what would determining this value tell us?

We do not exclude the possibility that long dsRNA is transmitted to the eggs alongside siRNAs resulting from Dicer processing in the mother. To characterize activity of the RNAi cellular machinery in the egg, ideally we would determine the relative contributions of transmitted siRNAs and transmitted long dsRNA molecules (that are subsequently processed) in achieving mRNA knockdown during embryogenesis. In theory, whether long dsRNA transmission is sufficient could be evaluated in a Dicer zygotic null background, where only transmitted siRNAs are present. However, we have yet to establish such a genetic model, and conversely a genetic approach to specifically block transmission of siRNAs remains elusive. These notions are newly encapsulated at the end of discussion section 3.1 (p. 14).

We do not understand how the reviewer’s suggestion of dosage experiments would address how much dsRNA is directly transmitted. Even if all knockdown in the embryo is achieved strictly through transmitted long dsRNA (and not by transmitted siRNAs), how would earlier waning or more rapid waning (two distinct possibilities) be

informative? Parental RNAi is transient: it wanes. And, while it is roughly proportional to the amount of dsRNA initially introduced into the mother, the scaling factors for this are entirely unknown, for a system that is probably saturated, and where dsRNA levels to achieve certain knockdown thresholds can be highly gene-specific. Also, dilution series do not necessarily give clean results.

For example, in Panfilio et al 2006, injected dsRNA at 10 µg, 1 µg, and 0.1 µg in the large milkweed bug adult female resulted in knockdown phenotypes in 46%, 86%, and 68% of offspring, respectively (n>100 embryos, doi:10.1016/j.ydbio.2005.12.028; cited ref. 60). For all three dsRNA amounts, the strongest phenotype possible was observed at those frequencies, but clearly there is no readily interpretable correlation between amount of dsRNA injected and resulting % affected offspring. In that paper, we wrote, “This might suggest that the dilution series did not use low enough concentrations to see a biological effect (especially compared to the degree of variation in uptake of dsRNA due to wound leakage).” Other dilution series, or variations in dsRNA length, have also been published by other groups. These collectively emphasize the importance of gene-specific thresholds of dsRNA for achieving stronger or weaker embryonic phenotypes, and that these thresholds can only be empirically determined through multiple experimental trials. Prior studies also show that strong and weak phenotypes can co-occur in synchronous clutch mates.

In the present study, to capture the waning time course (Figure 3), we ran multiple experiments for many months each. These are incredibly long-term and labor-intensive experiments. Doing an array of further such experiments to perhaps catch relative, gene-specific values would expand our dataset, but not substantially augment our findings that multiple parameters influence pRNAi time courses.

1.6 line 175. If wild type expression is negligible at 16-24 hours, how can there be efficient knockdown, do you see higher levels of mRNA to be knocked down? The results seem to be interpreted rather than described and then interpreted. The data reveal increased levels of RNA in RNAi individuals compared to wt, and only with probes internal to the dsRNA. This could be interpreted as coming from residual dsRNA. Similar results with multiple probes spanning the dsRNA suggest the residual RNA is likely full length.

A1.6) Apologies for the ambiguity. The reviewer is correct that peak expression of this gene occurs within 8-24 h, and it is not expressed (“negligible” levels detected by RT-qPCR) at 48-56 h and 64-72 h. The stage 16-24 h, a subset of 8-24 h, has very low but still detectable wild type mRNA expression. It is against this baseline that the RNAi samples were evaluated for the experiment presented in Figure 1C. To clarify, we now use the term “very low” instead of negligible in referring to the 16-24 h samples (results p. 6; Figure 1C legend p. 25). And yes, we interpret the expression levels from nested amplicon fragments 2-5 as representing dsRNA. Our comment, “the measured expression at this stage largely represents transmitted dsRNA present in the egg” builds on the findings and conclusions of the preceding section, including the newly added comment that “when wild type expression is low or undetectable, the dsRNA would constitute the majority or all detected RNA” (response A1.4.). This reflects the progression of the manuscript, but we are satisfied that the description of the experiment is factual and in keeping with preceding material (e.g., “efficient knockdown ... to ≤25% of wild type levels”, p. 6 second paragraph, is explained in the new prefatory comments on the ratio (R_{RNAi}/R_{WT}), as described in response A1.1).

1.7 Line 201 "overexpression" is an interpretation, while the observed result is higher levels of RNA in RNAi individuals.

A1.7) Please see our response A1.1 on the prior use of the term “overexpression”. In this instance we now use the phrase “more RNA in the RNAi than WT samples” (p. 7).

1.8 Line 406 "primer design may be constrained" is rather vague. It is hard to tell if this is a recommendation, observation of past experiments or problem that might be encountered. Please specify or clarify.

A1.8) This topic sentence introduces the specific details in the rest of the paragraph. To make this more explicit, we now preface these sentences with the phrases “for example”, “secondly”, and “thirdly” (p. 13).

1.9 Line 552 "Our unexpected finding that the long dsRNA molecule is maternally transmitted into eggs, thereby depleting maternal dsRNA levels, is difficult to reconcile with pRNAi persistence for months (Figs. 1-3)". This statement is difficult to accept. First they did not directly demonstrate that the maternal levels of dsRNA do actually decline as the pRNAi effect wanes. Second, what are the alternatives? If the dsRNA is processed prior to being imported into the embryos, wouldn't that also deplete the maternal levels of long dsRNA?

A1.9) We infer maternal long dsRNA depletion from evidence of complete waning of the knockdown effect over time (Figure 3), and the absence of any known mechanisms of long dsRNA amplification (third paragraph of discussion section 3.5, p. 18). Building on response A1.5, perhaps we should clarify that as developmental geneticists our working understanding is that, being systemic, RNAi of course occurs both in the mother and in offspring. (This is why studying embryonic phenotypes sometimes requires embryonic injection, to bypass maternal knockdown phenotypic consequences, such as in cited ref. 25 from our prior work [Horn and Panfilio 2016]. The overall experimental design for transgenerational knockdown, including maternal and zygotic effects, is also addressed in the large RNAi screening study in *Tribolium*: cited ref. 15 [Schmidt-Engel et al 2015].) So indeed long dsRNA processing and therefore dsRNA degradation occurs in the mother and will contribute to maternal dsRNA depletion. The direct transmission of dsRNA to offspring is a second route of depletion. Our intent with this sentence was to contrast finite amounts of dsRNA and its depletion with long-term knockdown. To clarify, we have changed “thereby depleting maternal dsRNA levels” to “contributing to depletion of maternal dsRNA levels” (p. 18).

We already mention “depletion of dsRNA from the mother” in the abstract and in the first paragraph of results section 2.4, including mention of dsRNA processing in the mother: “A single injection of the mother provides a finite number of dsRNA molecules, and the knockdown effect of pRNAi wanes over time in insects [1, 3, 36]. Our results suggest that waning may reflect not only endogenous transcript recovery after dsRNA degradation in the mother, but also maternal depletion of dsRNA due to its direct transmission into offspring” (p. 7).

These elements are further clarified within the discussion section (3.5) for which the sentence in question is the introduction. For example, we state, “the ovary represents just one organ in the female body in which dsRNA uptake occurs. In effect, the germline competes with other cell types for dsRNA. Particularly when it is distal to the site of dsRNA injection, it may be less sensitive or even refractory to RNAi [9, 23]” (second paragraph of section 3.5, p. 18).

1.10 Figure 6 is not referred to in the text. Relying on the figure legend for context, it appears to be showing the amounts of endogenous mRNA available to be knocked down at different time points. It is not clear how much dsRNA is present at these sample timepoints and how detection of it would affect the results of the assay. In particular, isn't there dsRNA present at the narrow sampling timepoint that obscures the true knockdown levels? And if the dsRNA is knocked down how can it "appear" at the longer sample timepoint? It would be interesting to see what the authors think is the relative amount of dsRNA present at these sample time points. Also please incorporate this figure and the data within in the main text.

A1.10) Figure 6 is indeed referred to in the text: within the Discussion section 3.1. The third paragraph of this section discusses developmental staging of endogenous gene expression profiles relative to the temporal precision of embryonic material used for RT-qPCR experiments. The reviewer is correct that the figure shows the amount of endogenous mRNA present (that could be knocked down) over developmental time.

It is the endogenous mRNA, not the dsRNA, that is measurably reduced after RNAi in our RT-qPCR experiments. We have no measure of long dsRNA levels in offspring independent of active embryonic RNAi machinery. However, please see response A1.5 regarding relative amounts of dsRNA and that the embryonic system appears to be saturated with dsRNA at constant levels. As we discuss in the main text (results section

2.1 final paragraph and discussion section 3.1 third paragraph), yes, presence of dsRNA leads to underestimation of knockdown strength when solely using a nested qPCR amplicon. This is still true when a narrow sampling timepoint is used, but the effect is then ameliorated. This is stated at the conclusions of the relevant main text paragraph: "Thus, staging precision is critical for accurate detection of knockdown efficiency, and this can largely overcome the underestimation effect of using a nested amplicon" (top of p. 14).

Building on the new paragraph in results section 2.1 that explains the RT-qPCR ratio R_{RNAi}/R_{WT} , in Discussion section 3.1 we have added a more general explication on this issue with respect to developmental staging and endogenous mRNA levels: this affects pp. 13-14 and a slight reworking of the Figure 6 legend on pp. 31-32 (including shifting a sentence to the main text).

1.11 The phylogenetic analysis is very thorough and convincing, although it only considers previously suggested candidate genes have largely been removed from consideration as candidate genes involved in the transmission of the dsRNA into the embryos.

A1.11) We agree that our phylogenomic profiling of candidate receptor proteins for cellular uptake is not comprehensive. Our goals were to thoroughly present our approach, including cautionary discussion on appropriate interpretation of orthology clustering data (discussion section 3.4), and to help lay to rest the persistent but erroneous focus on the "hot topic" protein SID-1 (section 3.3). We are pleased that the reviewer finds this analysis strategy convincing, and we hope that it will prove fruitful in wider application in future work.

Reviewer #2

Review of "Persistent parental RNAi in the beetle *Tribolium castaneum* involves maternal transmission of long double-stranded RNA" by Thorsten Horn, Kalin D. Narov and Kristen A. Panfilio for publication in *Advanced Genetics*.

The authors performed a trilateral investigation of the molecular mechanism of pRNAi in the model insect *Tribolium castaneum* that combines experimental results with comparative genomics. They present a RT-qPCR strategy that unmask the presence of dsRNA distinct from endogenous mRNA. This method revealed a strong pRNAi that can persist for months. In addition, the authors present phylogenomic profiling of possible receptor proteins for cellular uptake of long dsRNA into the egg.

The manuscript provides important basic knowledge on molecular mechanisms of dsRNA transmission during systemic pRNAi. The paper is very clear and nicely written, I only some minor comments, and suggestions (see below).

2.1 If the manuscript is not already on the maximum word count, I would suggest including more information in the 'Parental RNAi' section of the M & M (e.g. injected amount of dsRNA, transcription of dsRNA etc.). The authors refer to citation no 25, which is than referring to the manuscript of van der Zee et al. (2005). Referring from one publication to another is a bit annoying, including all the needed information in just one publication, if possible, is much more convenient for the reader.

A2.1) We agree that complete documentation in a single place is preferable. We now present the "RT-qPCR experiments" section before the "Parental RNAi" methods section to cross-reference common procedures, and we have augmented the account of dsRNA preparation (p. 21).

2.2 Is Line 135: 'semi-nested amplicon' - is this fragment 1?

A2.2) Yes, we have added parenthetical notes in the manuscript to clarify this (p. 5, first paragraph).

2.3 In line 136&137 the authors refer to an 'original assay'. Could the authors include a specific reference? I was not sure which study is meant, ref no 20? Ref no 25?

A2.3) Apologies for the ambiguity. Here we refer to experimental data solely within this study. We have replaced the term “original” with “initially-used”, in conjunction with citing the data in Figure 1 (p. 5, first paragraph).

2.4 The authors might consider including studies performed by the group of Palli about endosomal entrapment in the discussion e.g.: Yoon JS, Gurusamy D, Palli SR. (2017) doi: 10.1016/j.ibmb.2017.09.011

A2.4) We fully support citing primary studies where appropriate, but on reflection we have left the text as is. Our comments in the discussion on endosomal entrapment are purely based on our own survey of the literature, for the sake of conceptual rigor although we do not present primary data on this issue. For this reason, we emphasize that the image panel Figure 7D is based on multiple review articles (our references 4, 5, 42). Of these, reference 5 is a very recent review from the Palli lab that offers their own synthesis across multiple primary data papers from their lab’s ongoing work on this topic, specifically presented in detail in their figure 2 of the review article. We refer to this existing synthesis from the authors in the current text: “Species-specific levels of dsRNA sequestration have been correlated with susceptibility to RNAi [5]. ... Beetles including *Tribolium* appear to have low levels of endosomal sequestration, but those studies were performed in larvae [reviewed in 5].” (p. 19, second paragraph).

2.5 Figure 6 is missing the meaning of the abbreviation 'KD' in the figure legend.

A2.5) Apologies for the omission. We now explicitly indicate that “KD” means RNAi knockdown (p. 31).

2.6 Figure 7 looks a bit unorganized and to me it is not clear what is displayed in 7B.

A2.6) In the figure legend, we state, “Cartoons represent the progression of dsRNA from initial injection (**A**), through the mother’s tissues (**B**) and cells (**C,D**), to the oocytes (**E**).” To clarify, we have now added the additional text, “In detail, we depict injection into the abdominal body cavity (A); clearance from the hemolymph (B, schematized representation of the dorsal vessel (heart) and major circulatory system branching structures such as arteries, [91]); uptake across the plasma membrane into individual cells, which may be receptor-mediated (C); and potential sequestration from the cytosol in endosomes (D). Despite challenges associated with each of these steps (see Discussion section 3.5), systemic parental RNAi involves dsRNA is transmitted from the ovary into oocytes for months (E).” (p. 32). We acknowledge that the insect circulatory system is largely open and with segmentally iterated major anatomical features, unlike the fine capillaries in vertebrates and unlike the insect respiratory system, but we have tried to schematize the structure for primary distribution of hemolymph that also initially circulates dsRNA.

The authors have addressed the reviewers' comments and the revised manuscript is accepted for publication.